# Phase separation of Nur77 mediates celastrol-induced mitophagy by promoting the liquidity of p62/SQSTM1 condensates

Shuang-zhou Peng[1,3], Xiao-hui Chen[1,3], Si-jie Chen[1], Jie Zhang[1], Chuan-ying Wang[1], Wei-rong Liu[1], Duo Zhang[1], Ying Su[1,2] & Xiao-kun Zhang [1✉]

Liquid-liquid phase separation promotes the formation of membraneless condensates that mediate diverse cellular functions, including autophagy of misfolded proteins. However, how phase separation participates in autophagy of dysfunctional mitochondria (mitophagy) remains obscure. We previously discovered that nuclear receptor Nur77 (also called TR3, NGFI-B, or NR4A1) translocates from the nucleus to mitochondria to mediate celastrol-induced mitophagy through interaction with p62/SQSTM1. Here, we show that the ubiquitinated mitochondrial Nur77 forms membraneless condensates capable of sequestrating damaged mitochondria by interacting with the UBA domain of p62/SQSTM1. However, tethering clustered mitochondria to the autophagy machinery requires an additional interaction mediated by the N-terminal intrinsically disordered region (IDR) of Nur77 and the N-terminal PB1 domain of p62/SQSTM1, which confers Nur77-p62/SQSTM1 condensates with the magnitude and liquidity. Our results demonstrate how composite multivalent interaction between Nur77 and p62/SQSTM1 coordinates to sequester damaged mitochondria and to connect targeted cargo mitochondria for autophagy, providing mechanistic insight into mitophagy.

[1] School of Pharmaceutical Sciences, Fujian Provincial Key Laboratory of Innovative Drug Target Research, Xiamen University, Xiamen 361102, China. [2] NucMito Pharmaceuticals Co. Ltd., Xiamen 361101, China. [3] These authors contributed equally: Shuang-zhou Peng, Xiao-hui Chen. ✉email: xkzhang@xmu.edu.cn

Mitophagy, special selective autophagy with an evolutionarily conserved process that eliminates impaired mitochondria through autophagy, is important for cellular homeostasis and prevention of diseases and cancer[1–4]. The mechanism by which dysfunctional mitochondria are recognized and engulfed by autophagosomes involves ubiquitination of mitochondrial proteins followed by their interaction with cargo receptors connecting targeted cargo to the autophagosomal membrane via LC3/ATG8. Several cargo receptors including OPTN, NDP52, p62/SQSTM1 (hereafter referred to as p62), NBR1, and TAXBP1, are known to interact with both ubiquitin chains attached to damaged mitochondria and with LC3/ATG8 proteins anchored to the inner surface of the autophagosome membrane[5,6]. Different types of autophagy receptors are likely required for mitophagy depending on different physiological and pathological situations[3].

p62, one of the best characterized selective autophagy receptors, participates in different types of selective autophagy[7–11]. Like other cargo receptors, p62 contains a ubiquitin-associating (UBA) domain responsible for binding ubiquitinated proteins, and an LC3-interacting region (LIR), thus acting as a cargo receptor by simultaneously binding to the cargo and LC3[12]. In addition, p62 harbors a Phor and Bem1p (PB1) domain at its $NH_2$-terminal region, which mediates p62 oligomerization essential for its function as a selective autophagy receptor[13–16]. PB1-directed organization of p62 filamentous assemblies is a critical determinant of the diverse functions of p62. While p62 was initially shown to be required for PINK1/Parkin-mediated mitophagy[17], later studies revealed that the recruitment of p62 by ubiquitination of mitochondrial outer membrane proteins only serves to aggregate/cluster dysfunctional mitochondria through polymerization via its PB1 domain[18,19], in a manner analogous to its aggregation of polyubiquitinated proteins[8,9]. This suggested that p62 fails to function as a full autophagy receptor in Parkin-mediated mitophagy. Interestingly, p62 is essential for inflammasome NLRP3 agonist-induced mitophagy[20] and the elimination of damaged mitochondria through autophagy in leukemia cells[21], demonstrating that p62 is fully functional in mediating mitophagy under different cellular contexts. However, how p62-mediated sequestration of dysfunctional mitochondria and the recruitment of the autophagy machinery are coupled and regulated remains unclear.

Nur77 (also called TR3, NGFIB, or NR4A1), a nuclear receptor superfamily member and an immediate-early response gene, plays an integral role in a plethora of cellular processes including survival, apoptosis, autophagy, immunity, and inflammation in response to diverse stimuli[22–24]. Like other nuclear receptors, Nur77 comprises an $NH_2$-terminal intrinsically disordered region (IDR) with a poorly defined function, a central DNA-binding domain (DBD), and a COOH-terminal ligand-binding domain (LBD). Although nuclear receptors are conventionally considered as transcription factors, we reported that Nur77 could translocate from the nucleus to mitochondria to modulate mitochondrial activities[25–27]. Recently, we discovered that celastrol could bind Nur77 to induce Nur77 translocation from the nucleus to damaged mitochondria, where it mediates their elimination through autophagy by interacting with p62[28]. Celastrol is a potent anti-inflammatory triterpenoid quinine-methide isolated from the root of a traditional Chinese medicinal herb *Tripterygium wilfordii* commonly known as "Thunder God Vine" that has been widely and successfully used for treating a number of autoimmune and inflammatory diseases including rheumatoid arthritis and lupus[28–30]. We showed that the elimination of dysfunctional mitochondria by celastrol-induced mitophagy alleviates chronic inflammation in several disease models, revealing a critical role of Nur77/p62-mediated mitophagy in controlling mitochondrial homeostasis. However, how Nur77 interaction with p62 promotes the autophagy of dysfunctional mitochondria remains unknown.

Liquid-liquid phase separation (LLPS) is an important mechanism of how cells rapidly and reversibly compartmentalize their components into membraneless organelles to mediate diverse biological processes[31–36]. Recent studies have revealed an important role that phase separation plays in the condensation of misfolded and ubiquitin-positive proteins and their degradation by autophagy[37,38]. p62 undergoes phase separation upon binding to polyubiquitin chain, driving misfolded and ubiquitinated protein aggregates into larger condensates in a PB1-dependent manner and their subsequent tethering to the autophagosomal membrane via its interaction with LC3 proteins[16,39,40]. However, little is known about the role of phase separation in the sequestering and tethering of larger intracellular organelles such as dysfunctional mitochondria.

In this work, we show that Nur77 and p62 assemble condensates to mediate celastrol-induced mitophagy. Celastrol-induced ubiquitination in the C-terminal LBD of Nur77 and its interaction with p62 results in the formation of Nur77-LBD/p62 condensates capable of sequestering damaged mitochondria. However, such condensates are insufficient to tether the clustered mitochondria to the autophagosomal membrane. The Nur77/p62 condensates require an additional interaction mediated by the N-terminal IDR of Nur77 with the PB1 of p62, which confers the condensates with liquidity. These results reveal a critical role of coordinated multivalent interaction between Nur77 and p62 in celastrol-induced mitophagy and provide mechanistic insight into nuclear receptor action.

## Results

### Nur77 and p62 form condensates in celastrol-induced mitophagy. To study the role of phase separation in celastrol-induced mitophagy, we first used a sensitive fluorescence-activated cell sorting (FACS)-based dual-color fluorescence-quenching assay with an EGFP-mCherry-COX8 reporter localized to the mitochondrial matrix to monitor celastrol-induced mitophagy[41,42]. Normal mitochondria are yellow, having both green and red fluorescence in the matrix, whereas mitochondria engulfed into acidic lysosomes environment show red-only fluorescence due to the selective sensitivity of EGFP fluorescence to low pH. One advantage of such analysis allows the detection of mitochondrial proteins in lysosomes during mitophagy. Mitochondria are susceptible to inflammatory signaling, which can result in the accumulation of inflammatory mediators including tumor necrosis factor receptor-associated factor 2 (TRAF2) at dysfunctional mitochondria[28]. We showed previously that celastrol can induce Nur77 translocation from the nucleus to damaged mitochondria where it interacts with TRAF2 to initiate mitophagy for eliminating dysfunctional mitochonria[28]. Thus, cells were treated with TNFα and celastrol to evaluate celastrol-mediated mitophagy throughout the study. Upon treatment with celastrol and TNFα, significant amounts of mitochondria were engulfed by lysosomes as indicated by their red fluorescence (Fig. 1a). Quantification of red-only autophagic mitochondria by flow cytometry (Supplementary Fig. 1a, b) revealed a temporal effect of celastrol on the autophagy of dysfunctional mitochondria, which peaked between 3 to 6 h of treatment (Supplementary Fig. 1c). The celastrol-induced appearance of red-only mitochondria and degradation of cytochrome *c* oxidase subunit II (COXII), a mitochondrial inner membrane protein, was attenuated in cells transfected with Atg7 siRNA, which inhibited the expression of Atg7 protein (Supplementary Fig. 1d), thereby confirming the mitophagic effect of celastrol and the feasibility of using FACS-based EGFP-mCherry-COX8 mitophagy assay.

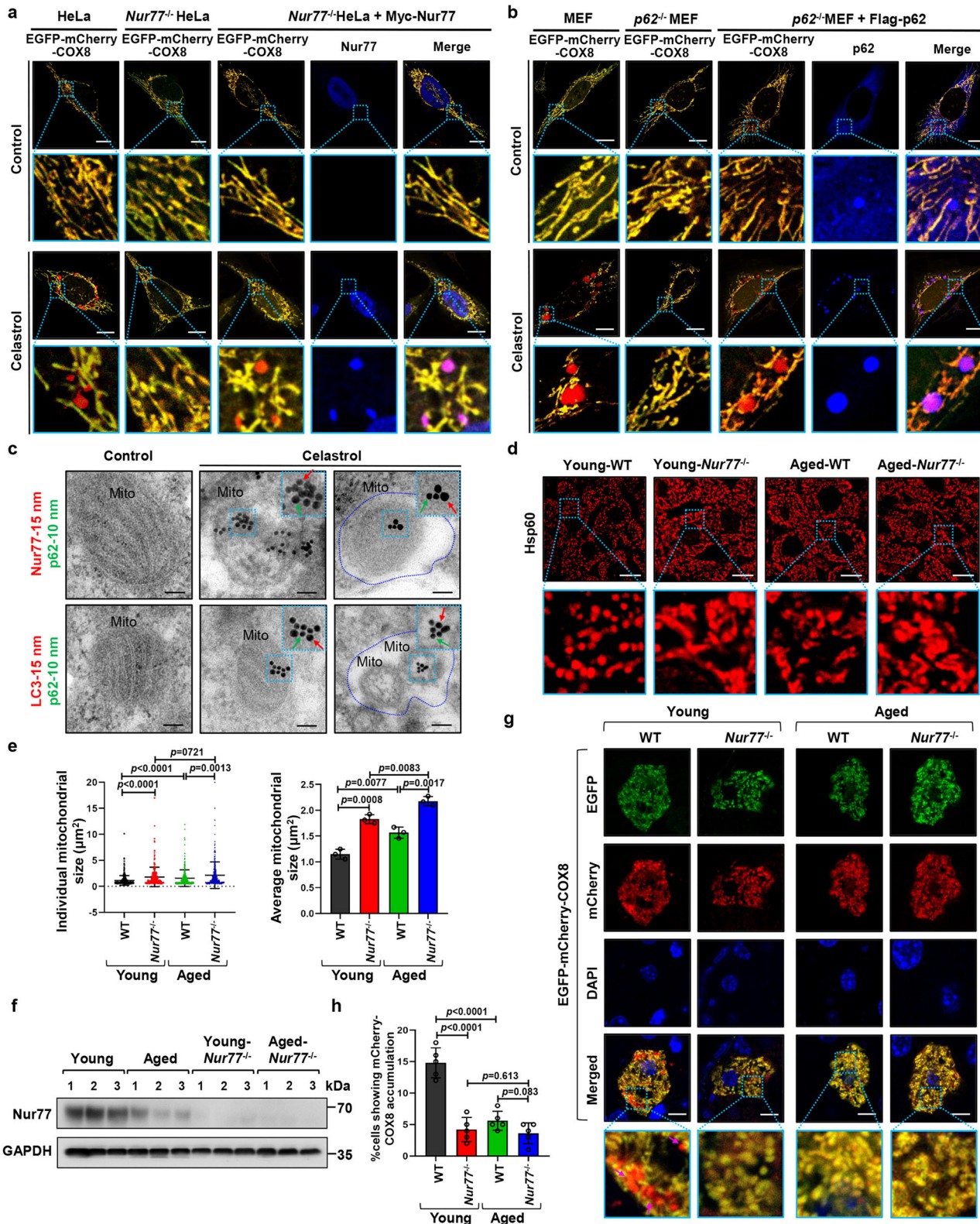

The role of Nur77 in celastrol-induced mitophagy was illustrated by reduction of autophagic mitochondria (from 12.20 to 0.02%) and diminished COXII degradation in HeLa cells depleted with Nur77 ($Nur77^{-/-}$HeLa) (Fig. 1a and Supplementary Fig. 1e). In addition, transfecting Myc-tagged Nur77 (Myc-Nur77) into $Nur77^{-/-}$HeLa rescued the effect of celastrol on inducing mitophagy (from 0.02 to 11.91%). Interestingly, Myc-Nur77 expressed in $Nur77^{-/-}$HeLa cells formed droplet-like condensates colocalizing with mitochondria undergoing autophagy in autolysosome (Fig. 1a), suggesting that Nur77 is directly involved in both sequestering dysfunctional mitochondria and directing cargo mitochondria for autophagy.

When the role of p62 was examined, we found that the mitophagic effect of celastrol was almost completely abolished in

**Fig. 1 Nur77 and p62 are required for celastrol-induced mitophagy. a** Representative image of celastrol-induced mitophagy in HeLa, $Nur77^{-/-}$HeLa, and $Nur77^{-/-}$HeLa cells transfected with Myc-Nur77 by EGFP-mCherry-COX8 assay as described in Methods. Scale bar, 10 μm. **b** Representative images of celastrol-induced mitophagy in MEFs and $p62^{-/-}$MEFs by EGFP-mCherry-COX8 assay as described in Methods. Scale bar, 10 μm. **c** Colocalization of Nur77, LC3, and p62 with mitochondria within mitophagosome/autolysosome. Upper panel: Electron micrographs of HeLa cells stained with 15 nm immunogold-conjugated Nur77 antibody to detect Nur77 (red), and 10 nm immunogold-conjugated p62 antibody to detect p62 (green). Bottom panel: Electron micrographs of HeLa cells stained with 15 nm immunogold-conjugated LC3 antibody to detect LC3, and 10 nm immunogold-conjugated p62 antibody to detect p62. Cells were treated for 1 h with celastrol. The blue dotted line indicates mitophagosome/autolysosome. Mito mitochondrion, Scale bar, 200 nm. **d** Representative images showing Hsp60, a mitochondrial marker, in the liver tissue from wild-type and $Nur77^{-/-}$mice in aging model. Young mice, 8 weeks old. Aged mice, 2 years old. Scale bar, 10 μm. **e** Statistical analysis of mitochondrial size was represented from liver tissue. Left graph, $n = 316$, 253, 267, and 287, respectively; Right graph, $n = 3$ biologically independent samples. A two-tailed unpaired Student's $t$-test was used for statistical analysis, and data were presented as mean values ± SEM. **f** The expression of Nur77 protein in the liver tissue from wild-type and $Nur77^{-/-}$mice in the aging model. **g** Representative images of EGFP-mCherry-COX8 in the liver from wild-type or $Nur77^{-/-}$mice in the aging model. Purple arrows indicate mitophagy. Scale bar, 2 μm. **h** Quantification of cells showing mCherry-COX8 accumulation on liver tissue. Two-tailed unpaired Student's $t$-test was used for statistical analysis, and data were presented as mean values ± SEM ($n = 5$ mice per group). Data represent at least three independent experiments. Source data are provided as a Source Data file.

$p62^{-/-}$MEF cells, which could be restored by reexpressing Flag-tagged p62 (Flag-p62) (Fig. 1b). Knocking down p62 from HeLa cells also impaired the mitophagic effect of celastrol (Supplementary Fig. 1f). Thus, p62 is also essential for celastrol-induced mitophagy. Interestingly, transfected Flag-p62, like Nur77 (Fig. 1a), was found in autolysosomes as droplet-like condensates colocalizing with autophagic mitochondria (Fig. 1b). Moreover, the condensates of Nur77 and p62 colocalized not only with each other but also with mitochondria in cells treated with celastrol (Supplementary Fig. 1g). The role of p62 in celastrol-induced mitophagy differs from Parkin-mediated mitophagy, in which p62 failed to translocate to autolysosome in oligomycin/antimycin A (OA)-induced mitophagy (Supplementary Fig. 1h), in agreement with previous observations[18,19]. Thus, p62 acts as a full selective autophagy cargo receptor for Nur77- but not Parkin-mediated mitophagy.

We also conducted immunogold electron microscopy (EM)[43] to examine the localization of Nur77 and p62 condensates. Dual immunogold staining revealed no detectable Nur77 or p62 on healthy mitochondria in control cells. When cells were treated with celastrol, damaged mitochondria were densely decorated at their outer membrane with anti-p62 (10 nm) and anti-Nur77 (15 nm) immunogold labels, which could be found within mitophagosome or autolysosome (Fig. 1c). LC3 (15 nm) and p62 (10 nm) immunogold particles were also colocalized near the mitochondrial outer membrane, which again were engulfed by the mitophagosome or autolysosome. These results support the notion that Nur77/p62 condensates are capable of directing cargo mitochondria to the autophagy machinery for degradation.

**Nur77-mediated mitophagy in aging**. The physiological significance of Nur77/p62-mediated mitophagy was examined during the aging process of the liver, in which the autophagy of damaged mitochondria plays a critical role in counterbalancing age-related pathological conditions[44]. Confocal microscopy analysis revealed that the size of mitochondria in the liver of aged mice significantly increased (Fig. 1d, e), in agreement with the enlargement of mitochondria termed megamitochondria in aged liver[45]. The increase in the size of mitochondria in aged mice was accompanied with liver enlargement (Supplementary Fig. 1i, j) and increased hepatic inflammation (Supplementary Fig. 1k). The role of Nur77 was revealed by enlarged hepatic mitochondria in both young and aged $Nur77^{-/-}$mice compared to wild-type mice (Fig. 1d, e) and decreased expression of Nur77 during aging (Fig. 1f). Alteration of p62-mediated Parkin-independent mitophagy is responsible for the formation of megamitochondria in the liver[46]. We, therefore, studied whether Nur77 and p62 could mediate mitophagy during aging, and found their colocalization

with mitochondria and LC3 in the liver of young mice, which decreased in the liver of aged mice (Supplementary Fig. 1l–p). Thus, Nur77 and p62 are involved in mitophagy of the liver, which declines during aging. This was confirmed by our data showing that knocking out Nur77 caused defective mitophagy in the liver of both young and aged mice (Supplementary Fig. 1o, p). To provide more direct evidence supporting the role of Nur77 in mitophagy during aging, we injected adeno-associated virus serotype 9 (AAV-9) plasmid that expresses the mitophagy biosensor EGFP-mCherry-COX8[42] into the livers of young and aged mice. Two weeks after injection, mCherry-COX8 exposure was detected in the liver of young mice but not young $Nur77^{-/-}$mice (Fig. 1g, h, and Supplementary Fig. 1q), demonstrating that Nur77 is required for hepatic mitophagy of young mice. Hepatic mitophagy declined during aging as the mCherry-COX8 exposure was not detected in the liver of aged wild-type and $Nur77^{-/-}$mice. Thus, Nur77/p62-mediated mitophagy plays a role in maintaining mitochondria homeostasis in the liver.

**Celastrol promotes phase separation of Nur77/p62**. p62 is known to phase separate ubiquitinated proteins into condensates that become substrates for autophagy[16,40]. The fact that not only Nur77 but also p62 displayed punctate structures colocalizing with damaged mitochondria undergoing autophagy in autolysosome (Fig. 1) suggested that the phase separation of Nur77 and p62 are involved in celastrol-induced mitophagy. To test our hypothesis, HeLa cells ectopically expressing Nur77 fused with green fluorescent protein (GFP-Nur77) and mCherry-fused p62 (mCherry-p62) were examined for their phase separation in the absence or presence of celastrol. Transfected GFP-Nur77 resided exclusively in the nucleus, while mCherry-p62 was diffusely distributed in the cytoplasm. However, when cells were treated with celastrol, GFP-Nur77 formed cytoplasmic droplets colocalizing extensively with mCherry-p62 in a time dependent manner (Fig. 2a). To understand how GFP-Nur77/mCherry-p62 condensates are formed in the cytoplasm upon celastrol treatment, cells were treated with celastrol for 1 h and subjected to real-time microscopy visualization of the formation of the GFP-Nur77/mCherry-p62 condensates. We found that transfected GFP-Nur77 and mCherry-p62 rapidly formed large cytoplasmic bodies through the fusion of undetectable micro-sized droplets or detectable droplets in response to celastrol treatment (Fig. 2b, Supplementary Fig. 2a, b, and Supplementary Movies 1–3). Thus, celastrol induces the formation of Nur77/p62 bodies by triggering their phase separation. The autophagosome marker LC3 is recruited to p62 bodies after p62 undergoes phase separation through ubiquitin-induced oligomerization[16,40], which serves as a mechanism for linking p62-containing protein aggregates to

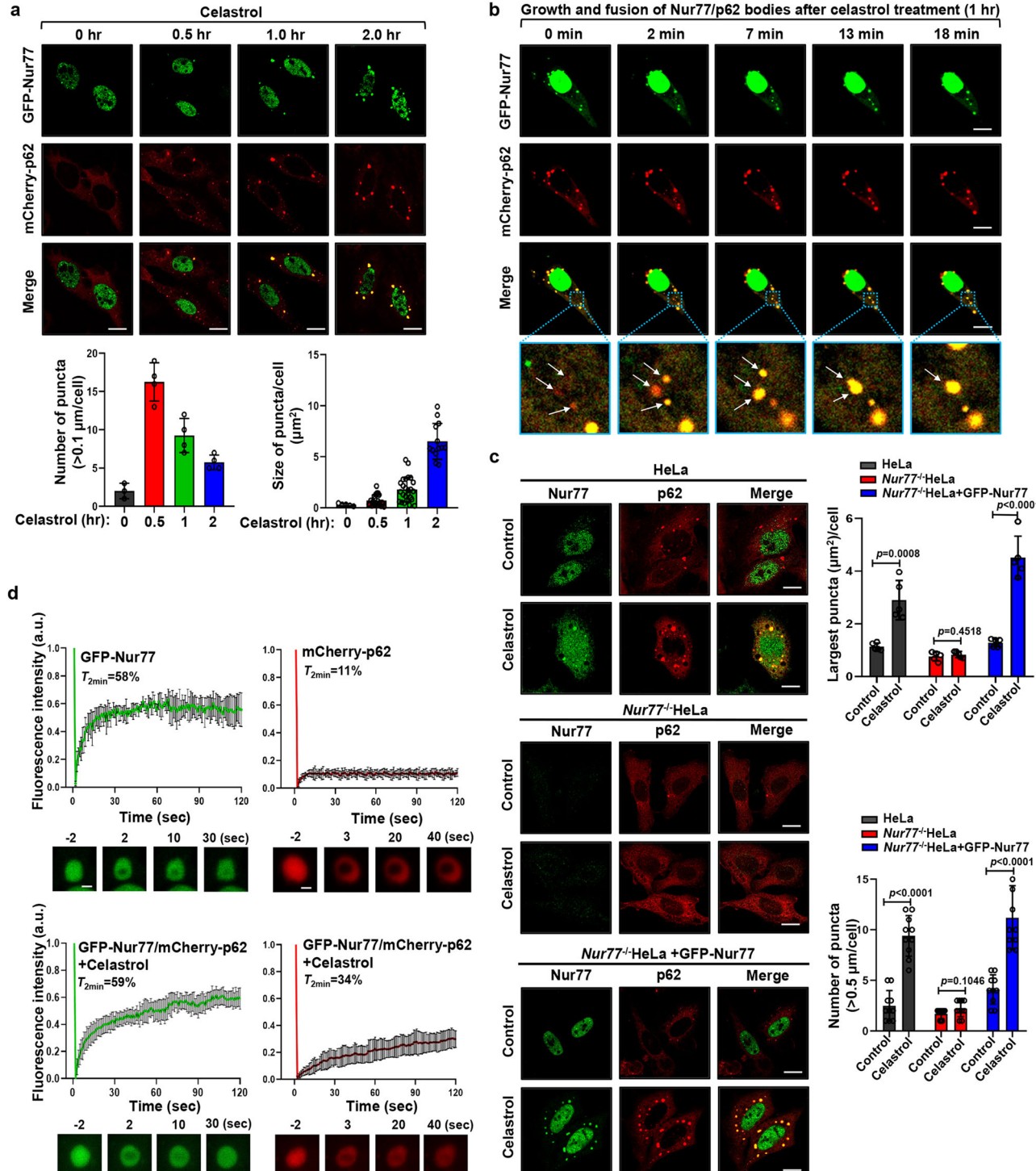

**Fig. 2 Celastrol promotes phase separation and liquidity of p62. a** Representative images showing the time-dependent effect on celastrol induction of cytoplasmic Nur77 body formation. Bottom panels: quantitative analysis of the number and size of Nur77/p62 body formation. Bottom left graph, $n = 4$ biologically independent samples; Bottom right graph, $n = 20$, 23, 25, and 19, respectively. Data were presented as mean values ± SEM. Scale bar, 10 μm. **b** Real-time images showing the formation and fusion of GFP-Nur77 and mCherry-p62 droplets in HeLa cells after treatment with celastrol (2 μM) for 1 h. White arrows indicate droplets formation and fusion (see also Supplementary Movie 1). Scale bar, 10 μm. **c** Representative images illustrating the role of celastrol in promoting p62 body formation in a Nur77-dependent manner immunostaining. $Nur77^{-/-}$HeLa cells were also transfected with GFP-Nur77 to determine its effect on p62 body formation. The diameter of the biggest p62 puncta in each cell was measured. The number of p62 puncta >0.5 μm in each cell was assessed. A two-tailed unpaired Student's $t$-test was used for statistical analysis, and data are presented as mean values ± SEM ($n = 5$ biologically independent samples). **d** FRAP analysis of the effect of Nur77 in regulating p62 mobility in HeLa cells. Data were presented as means ± SEM ($n = 3$ independent experiments). Scale bar, 1.5 μm. Data represent at least three independent experiments. Source data are provided as a Source Data file.

the autophagy machinery. However, whether LC3 can be recruited to phase-separated p62 bodies during mitophagy is unknown. We found that both LC3 and p62 localized at the outer membrane of mitochondria undergoing autophagy in the mitophagosome (Fig. 1c). Transfected Nur77 and p62 also formed condensates with LC3 and mitochondria in cells treated with celastrol (Supplementary Fig. 2c, d). These results confirmed the capability of Nur77/p62 condensates in mediating celastrol-induced mitophagy.

**Celastrol-induced phase separation of p62 depends on Nur77.** To determine the role of Nur77 in p62 phase separation, we studied the formation of p62 condensates in HeLa and *Nur77−/−*HeLa cells. In response to celastrol, p62 rapidly formed condensates colocalizing with Nur77 in HeLa cells (Fig. 2c). However, the ability of p62 to assemble condensates was impaired in *Nur77−/−*HeLa cells, which was restored by reexpressing GFP-Nur77. Thus, celastrol-induced formation of p62 condensates is dependent on Nur77 expression. As the liquidity of condensate is a critical determinant for its selective autophagy[47,48], we studied the effect of celastrol on the liquidity of Nur77/p62 condensates. Fluorescence recovery after photobleaching (FRAP) analysis of co-transfected GFP-Nur77 and mCherry-p62 revealed that the fluorescence of GFP-Nur77 after photobleaching was recovered rapidly and significantly ($T_{2 \text{ min}} = 58\%$), while the fluorescence of p62 was only slightly recovered ($T_{2 \text{ min}} = 11\%$) and remained static (Fig. 2d), indicating a more liquid-like property of GFP-Nur77 than mCherry-p62. This is consistent with previous studies showing low mobility of p62 body[16,40]. However, when cells were treated with celastrol, which induces the formation of Nur77/p62 bodies, the recovery rate of the p62 signal ($T_{2 \text{ min}} = 34\%$) was much improved, while the recovery of Nur77 in Nur77/p62 bodies remained almost the same. These results demonstrated that Nur77 is a critical determinant of the liquidity and mobility of the Nur77/p62 condensates.

**Nur77 forms liquid-like condensate through phase separation.** The above FRAP data suggested that Nur77 might itself undergo phase separation. Thus, GFP-Nur77 was expressed and purified to study its phase separation. When added to droplet formation buffer containing a crowding agent (10% PEG-3.35 K) to simulate the densely crowded environment of the nucleus, GFP-Nur77 but not GFP formed micro-sized droplets in solution with markedly increased size (Fig. 3a). FRAP analysis demonstrated that GFP-Nur77 redistributed rapidly from the unbleached area to the bleached area (Fig. 3b). GFP-Nur77 droplets also fused to form larger droplets, revealing droplet coalescence (Fig. 3c). The formation of Nur77 droplets was a general result of macromolecular crowding rather than a specific effect of PEG, as Nur77 assembled into spherical droplets in the presence of Dextran, Ficoll-400, or in a highly concentrated solution of lysozyme (2 mM) (Supplementary Fig. 3a). It was dependent on the molecular weight of the crowding agent, illustrated by its assembling into spherical droplets only in solutions containing >1 kDa PEG. The size of mCherry-Nur77 droplets in solution became bigger by increasing PEG molecular mass with the efficiency of condensate formation peaked in 8 kDa PEG (Supplementary Fig. 3b). Analysis of the biophysical properties of Nur77 droplets revealed a shift of the distributions and opacity of the droplets toward greater droplets with increasing PEG concentration (Supplementary Fig. 3c). Phase-separated droplets typically scale in size according to the concentration of components in the system[49]. Indeed, the size distribution and turbidity of Nur77 droplets increased in a Nur77 concentration-dependent manner (Supplementary Fig. 3d). We also tested whether the droplets were irreversible aggregates or reversible phase-separated condensates. To this end, GFP-Nur77 was allowed to form droplets in an initial solution, and then diluted sequentially by five times in equimolar protein concentration and crowding agent. The preformed GFP-Nur77 droplets were reduced in size and turbidity with dilution, demonstrating that the droplet formation was reversible (Supplementary Fig. 3e). Taken together, these results demonstrated that Nur77 exhibits liquid-like properties in vitro.

To test whether Nur77 undergoes LLPS in vivo, we ectopically expressed GFP-Nur77 in HeLa cells. Transfected GFP-Nur77 formed punctum structures in the nucleus of cells (Fig. 3d), in which GFP-Nur77 diffused rapidly within the puncta (Fig. 3e). 3D-reconstruction of the GFP-Nur77 images revealed that it exhibited as a single particle (Fig. 3f). Myc-Nur77 (Fig. 3g) and endogenous Nur77 (Fig. 3h) also formed puncta in the nucleus. Thus, the Nur77 phase separates into condensates via LLPS.

**Nur77 phase separation is dependent on its N-terminal IDR domain.** Proteins containing large IDRs often phase separate under physiologic conditions[50]. Using the prediction program PONDR (VSL2) (http://www.pondr.com/)[51], we found that the N-terminal region of Nur77 displays the highest structural disorder propensity (Fig. 4a). To determine if the IDR of Nur77 is required for its phase separation, GFP-tagged IDR (GFP-Nur77-IDR) or LBD of Nur77 (GFP-Nur77-LBD) (Fig. 4b) was purified and subjected to droplet formation in vitro. GFP-Nur77-IDR formed numerous spherical droplets in a buffer containing 10% PEG-3.35 K (Fig. 4c). For comparison, GFP-Nur77-LBD exhibited amorphous condensates rather than spherical droplets. GFP-Nur77-IDR also formed phase-separated droplets in cells (Fig. 4d, e). Removing IDR from Nur77 (GFP-Nur77-ΔIDR) significantly attenuated its ability to form nuclear droplets, while deleting DBD (GFP-Nur77-ΔDBD) altered the morphology Nur77 bodies. GFP-Nur77-LBD also failed to assemble droplets in cells. Thus, the N-terminal IDR of Nur77 is responsible for Nur77 phase separation.

The N-terminal IDR of Nur77 is involved in an intramolecular interaction with its carboxyl-terminal LBD[52]. We then asked whether GFP-Nur77-IDR could modulate the phase separation of mCherry-Nur77-LBD through the intramolecular interaction. Nur77-LBD alone exhibited amorphous condensates rather than spherical droplets (Fig. 4c). However, when mCherry-Nur77-LBD was mixed with GFP-Nur77-IDR, mCherry-Nur77-LBD formed spherical droplets together with GFP-Nur77-IDR (Supplementary Fig. 4a). For comparison, GFP-Nur77-IDR had no effect on droplet formation of mCherry and mCherry-DBD (Supplementary Fig. 4b). These studies revealed the potency of Nur77-IDR in LLPS.

Nuclear receptor superfamily members are characterized by the presence of IDRs in their N-terminal regions, whose function remains poorly defined. To address whether IDRs in other nuclear receptor family members could also mediate phase separation, we studied Nur77 subfamily members NOR1 and Nurr1. Similar to Nur77, NOR1, and Nurr1 contain a large IDR in their N-terminal region (Supplementary Fig. 4c, d). We then tested whether Nurr1 and NOR1 underwent LLPS in vivo and found that ectopically expressed GFP-Nurr1 and GFP-NOR1 also formed punctate structures in the nucleus (Supplementary Fig. 4e, f). Removal of IDR from Nurr1 (GFP-Nurr1-ΔIDR) or NOR1 (GFP-NOR1-ΔIDR) greatly reduced their ability to form nuclear droplets. GFP-Nurr1-LBD and GFP-NOR1-LBD also failed to form droplets. Thus, the N-terminal IDR of Nurr1 and NOR1 also mediates their formation of phase-separated droplets, suggesting that the IDR in the nuclear receptor superfamily members plays an important role in regulating nuclear receptor action through phase separation.

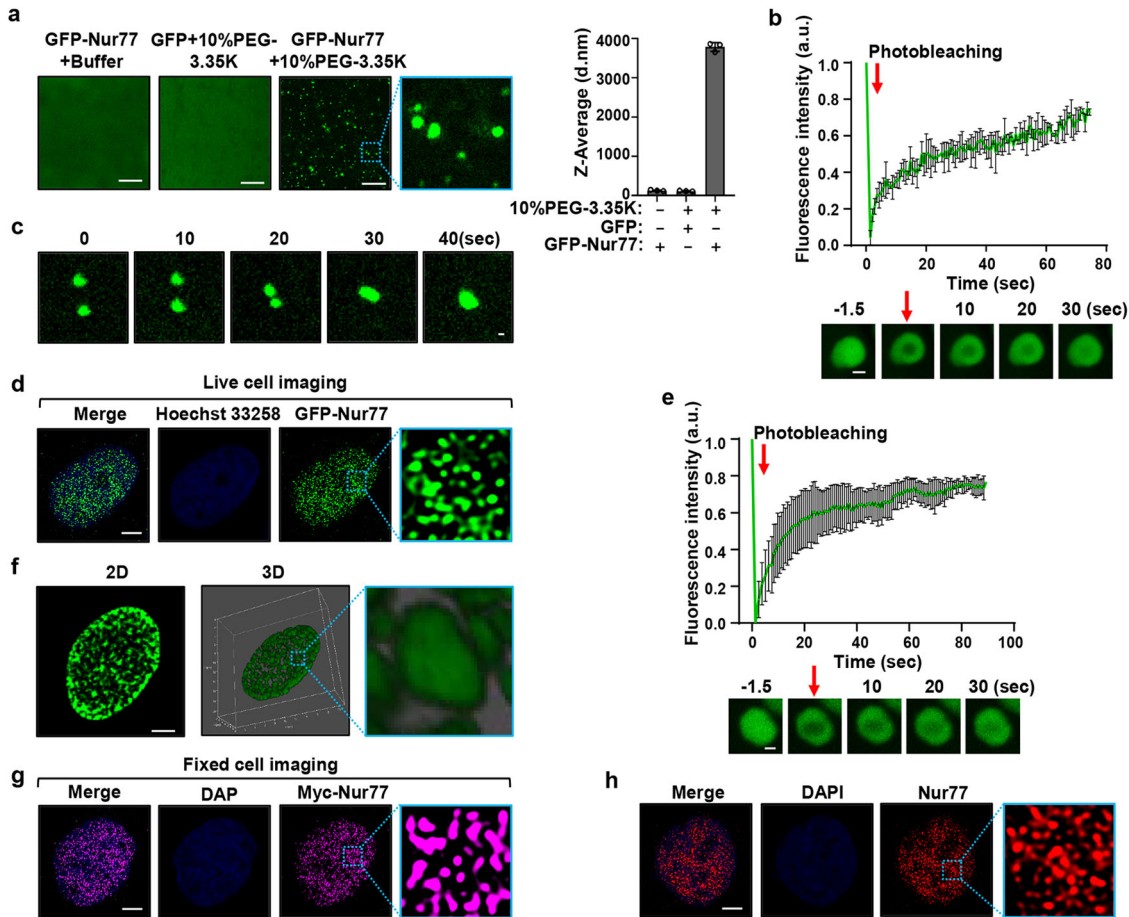

**Fig. 3 Nur77 phase separates into a liquid-like condensate. a** GFP-Nur77 (2 μM) undergoes phase separation. The size of Nur77 droplets was analyzed. Data were presented as mean values ± SEM ($n = 3$ independent experiments). Scale bar, 10 μm. **b** Top, changes in fluorescence intensity of GFP-Nur77 droplets after photobleaching were plotted over time. Bottom, representative images of fluorescence recovery. Data were presented as mean values ± SEM ($n = 3$ independent experiments). Scale bar, 1.5 μm. **c** Fusion of GFP-Nur77 droplets in 10% PEG-3.35 K. Scale bar, 20 μm. **d** Live imaging of GFP-Nur77 in HeLa cells. Scale bar, 5 μm. **e** Time course analysis of GFP-Nur77 nuclear body recovery after photobleaching in HeLa cells. Representative images of fluorescence recovery are shown. Data were presented as mean values ± SEM ($n = 3$ independent experiments). Scale bar, 1.5 μm. **f** Three-dimensional (3D) images of Nur77 nuclear assemblies. An enlarged view of inset is also shown. Scale bar, 5 μm. **g** Fixed imaging of Myc-Nur77 in HeLa cells. Scale bar, 5 μm. **h** Endogenous Nur77 displays nuclear puncta in HeLa cells revealed by immunostaining with anti-Nur77. Scale bar, 5 μm. Data represent at least three independent experiments. Source data are provided as a Source Data file.

**Celastrol regulation of Nur77 phase separation**. We next studied how celastrol binding to Nur77 regulates its phase separation. Transfected GFP-Nur77 exclusively resided in the nucleus of control cells (Figs. 1a and 2a). Figure 4f showed stills of movie taken after GFP-Nur77-transfected cells were treated with celastrol for 1 h (Supplementary Movie 4). There was a significant amount of cytoplasmic GFP-Nur77 bodies in cells with new bodies continuously emerging from the nuclear membrane. Thus, celastrol promotes GFP-Nur77 nuclear export as we reported before[26], although we could not exclude the possibility that celastrol may also promote the retention of cytoplasmic Nur77. Our data also showed the growth of Nur77 droplets in the cytoplasm via fusion, revealing a role of phase separation in celastrol-induced growth of cytoplasmic Nur77 bodies. When the effect of celastrol on phase separation of Nur77 mutants was analyzed (Fig. 4g), we found that celastrol could also facilitate the conversion of Nur77 mutants, Nur77-ΔIDR, and Nur77-LBD, from diffusely status to spherical particle structures in the nucleus, suggesting a ligand-dependent phase separation in the nucleus. The effect of celastrol is likely due to its binding to Nur77 as BI1071, another Nur77 ligand[53], also promoted Nur77 phase separation, while K-80003, a RXRα ligand[54,55], had no

effect. We also used FRAP analysis to study the mobility and liquidity of GFP-Nur77 and mutants and found that the fluorescence recovery of GFP-Nur77 and GFP-Nur77-ΔDBD was much quicker than that of GFP-Nur77-ΔIDR and GFP-Nur77-LBD (Supplementary Fig. 4g), revealing a critical role of Nur77-IDR in promoting the mobility and liquidity of celastrol-induced Nur77 condensates.

**Nur77 ubiquitination-induced p62 phase separation sequesters mitochondria**. Ubiquitinated substrates can trigger p62-mediated phase separation and segregation of the substrates into autophagic degradation[16,39,40]. We previously reported[28] that celastrol binding to Nur77 induces Nur77 translocation to mitochondria where it is ubiquitinated at Lys536 located in its C-terminal LBD by TRAF2, an E3 ubiquitin ligase[56], through interaction. The ubiquitinated mitochondrial Nur77 then interacts with the UBA of p62, leading to autophagy of Nur77-primed dysfunctional mitochondria[28]. To study the molecular detail of phase separation in mediating the mitophagic effect of Nur77 and p62, we examined the role of Nur77 ubiquitination in p62 phase separation. Consistent with our previous results[28], TRAF2 was recruited to the surface of Nur77 condensates in cells treated with celastrol,

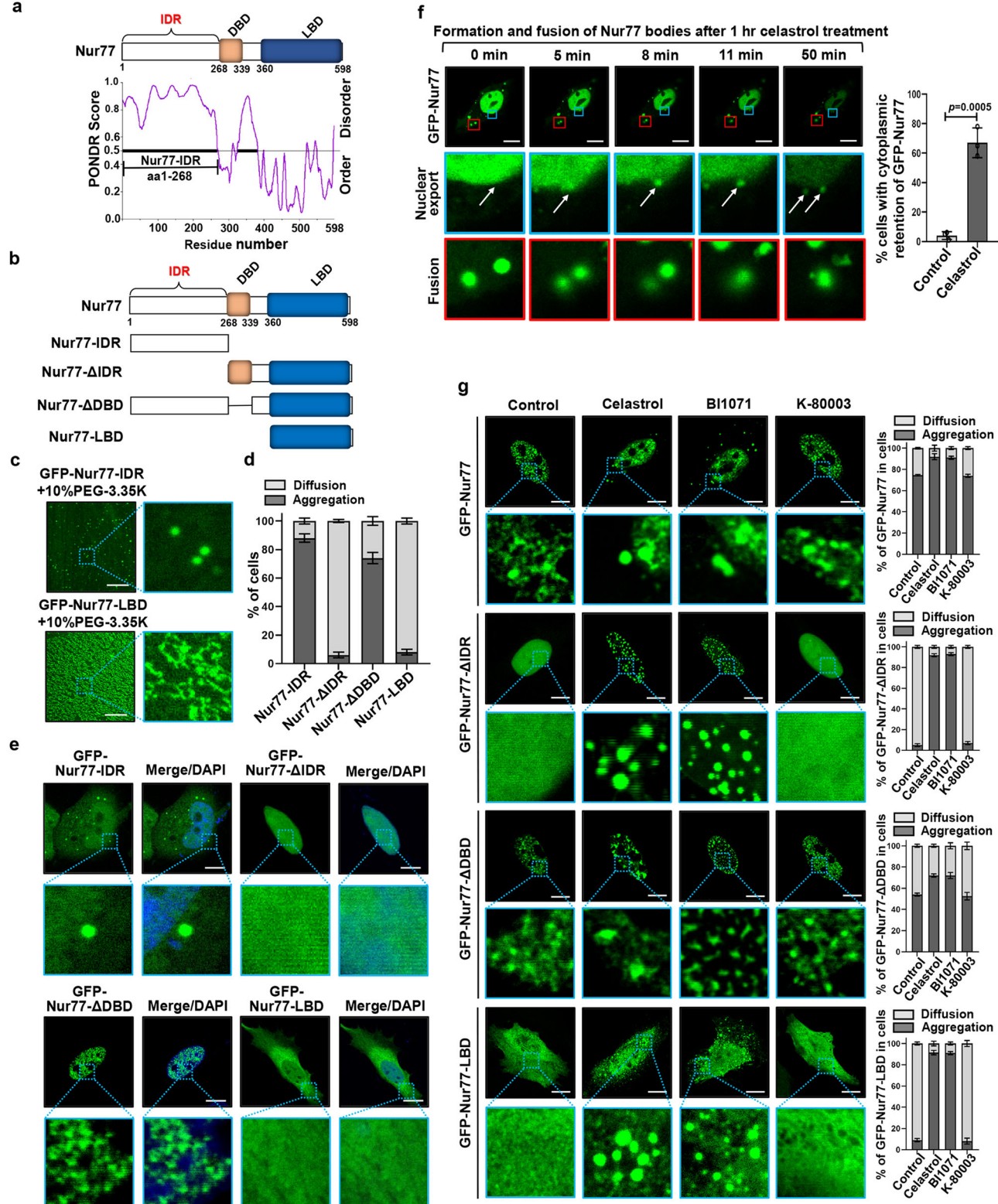

displaying spherical shell structure (Supplementary Fig. 5a). TRAF2-mediated Nur77 ubiquitination likely plays a role in p62 body formation as celastrol-induced Nur77 condensates colocalized not only with p62 but also with ubiquitin (Fig. 5a). To confirm the role of Nur77 ubiquitination, we mutated its ubiquitination site and tested its effect on p62 body formation. Mutating Lys536 in Nur77 (Nur77-K536R) (Fig. 5b) or Nur77-LBD (Nur77-LBD-K536R) (Fig. 5c) impaired the effect of

celastrol on inducing their interaction with p62. To our surprise, mutating Lys536 in Nur77 did not show an apparent inhibitory effect on the celastrol-induced assembly of Nur77/p62 condensates (Fig. 5d). We suspected that this might be due to celastrol-independent interaction between Nur77 and p62, which was not affected by Lys536 mutation (Fig. 5b). Notably, mutating Lys536 in Nur77-LBD completely impaired its ability to interact with p62 even in the presence of celastrol (Fig. 5c), indicating that

**Fig. 4 Nur77 phase separation is dependent on N-terminal IDR domain. a** Intrinsic disorder tendency of Nur77. IDR intrinsically disordered region, DBD DNA-binding domain, LBD ligand-binding domain. **b** Schematic representation of Nur77 and its mutants. **c** In vitro phase separation of GFP-Nur77-IDR and GFP-Nur77-LBD (2 μM). Scale bar, 10 μm. **d** Quantification of Nur77 mutant droplets formation in absence of celastrol in HeLa cells. Data were presented as mean values ± SEM (n = 3 independent experiments). **e** Representative images of Nur77 mutant droplets formation in absence of celastrol in HeLa cells. Scale bar, 10 μm. **f** Representative real-time images showing the formation and fusion of cytoplasmic GFP-Nur77 droplets after treatment with celastrol (2 μM) for 1 h in HeLa cells (see also Supplementary Movie 4). Right: quantification of the cytoplasmic retention of GFP-Nur77 protein. A two-tailed unpaired Student's t-test was used for statistical analysis, and data were presented as mean values ± SEM (n = 3 independent experiments). Scale bar, 10 μm. **g** Droplet formation of GFP-Nur77 and mutants in HeLa cells treated with the indicated compounds (2 μM). Left: Representative droplet images of transfected GFP-Nur77 and mutants. An enlarged view of the inset is also shown. Scale bar, 10 μm. Right: Quantification of droplet formation of GFP-Nur77 and mutants. Data represent at least three independent experiments. Source data are provided as Source Data file.

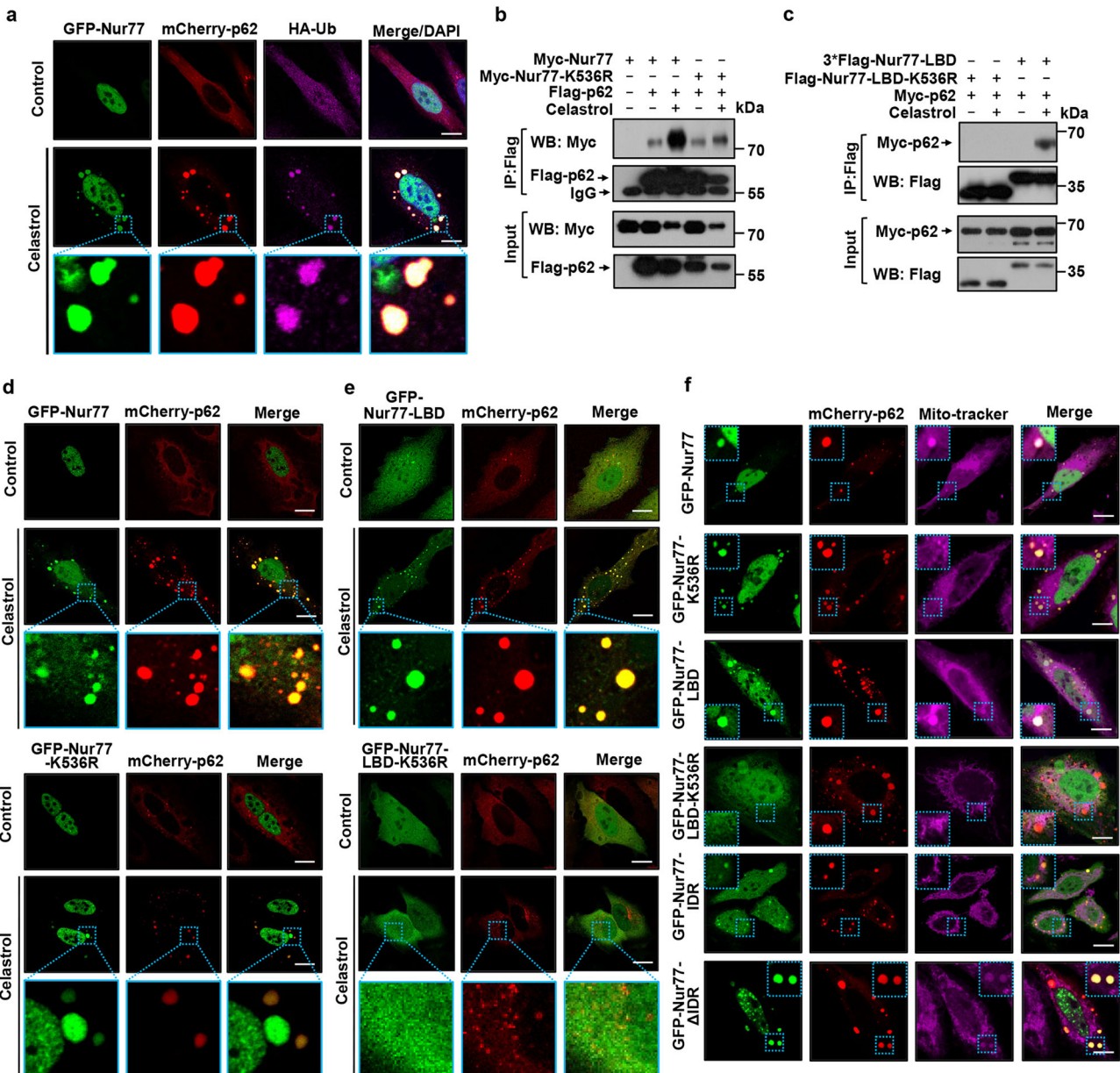

**Fig. 5 Ubiquitinated Nur77 interacts with p62 to sequester damaged mitochondria. a** Immunofluorescence images of GFP-Nur77, mCherry-p62, and HA-Ub transfected in HeLa cells treated with or without celastrol. Data illustrate the colocalization of Nur77 with p62 and Ub in the presence of celastrol. Scale bar, 10 μm. **b**, **c** Interaction of indicated Nur77 or deubiquitinated mutant (K536R) and p62 was analyzed in HeLa cells treated with or without celastrol by co-immunoprecipitation (co-IP) assay. **d**, **e** Immunofluorescence images showing the effect of celastrol-induced Nur77 ubiquitination on mCherry-p62 droplet formation. Scale bar, 10 μm. **f** Representative images showing ubiquitination-dependent colocalization of Nur77 with p62 and mitochondria in HeLa cells treated with celastrol. Scale bar, 10 μm. Data represent at least three independent experiments. Source data are provided as Source Data file.

Nur77-LBD is not involved in celastrol-independent interaction between Nur77 and p62. Thus, we used Nur77-LBD to study the effect of Nur77 ubiquitination on p62 body formation. Nur77-LBD could assemble condensates with p62 in the presence of celastrol. However, such an effect of Nur77-LBD was completely abolished when Lys536 was mutated (Fig. 5e). These results, therefore, revealed not only a critical role of celastrol-induced Nur77 ubiquitination in promoting p62 phase separation but also the existence of the celastrol-independent interaction between Nur77 and p62.

We next studied the role of Nur77 ubiquitination-mediated p62 phase separation in celastrol-induced mitophagy. Mitophagy involves sequestering/clustering dysfunctional mitochondria and subsequently directing cargo mitochondria for degradation by the autolysosome system[57]. We first studied its effect on sequestering dysfunctional mitochondria. Upon celastrol treatment, condensates formed by GFP-Nur77 and mCherry-p62 colocalized with mitochondria (Fig. 5f and Supplementary Fig. 5b). By contrast, the condensates assembled by Nur77-K536R and p62 did not. Thus, p62 recruitment by ubiquitinated Nur77 could mediate celastrol-induced sequestering of dysfunctional mitochondria, in a manner analogous to its aggregation of misfolded proteins[16,40] or dysfunctional mitochondria during Parkin-mediated mitophagy[18,19]. This is further confirmed by data showing that celastrol-induced Nur77-LBD/p62 but not Nur77-LBD-K536R/p62 condensates colocalized with mitochondria. As deleting IDR from Nur77 (Nur77-ΔIDR) did not affect its ability to aggregate mitochondria, the C-terminal LBD of Nur77 is sufficient to sequester damaged mitochondria in a ubiquitination dependent manner (Fig. 5f and Supplementary Fig. 5b). Notably, the IDR of Nur77 could also assemble condensates with mCherry-p62 even though it is not required for clustering mitochondria.

**Nur77 ubiquitination-mediated p62 phase separation is insufficient to complete mitophagy.** We next asked whether Nur77 ubiquitination-induced p62 phase separation is sufficient to complete the mitophagic process, in analogous to its effect on misfolded proteins[9]. We reasoned that if it was sufficient, Nur77-LBD, which promotes p62 phase separation through ubiquitination, should also act to tether clustered mitochondria to autolysosome for degradation. Bodies assembled by cotransfected GFP-Nur77 and mCherry-p62 were found in lysosome as indicated by their colocalization with Lyso-Tracker (Fig. 6a, b). GFP-Nur77 also colocalized with both mitochondria and lysosome (Fig. 6c, d). Thus, condensates containing Nur77, p62, and mitochondria were engulfed by autolysosome. The role of Myc-Nur77 to support the mitophagic effect of celastrol was confirmed by EGFP-mCherry-COX8 mitophagy assay and its reduction of COXII expression (Supplementary Fig. 6a, b). However, when GFP-Nur77-LBD was evaluated, we found that the condensates formed by GFP-Nur77-LBD and p62 were not detected in lysosome (Fig. 6a, b), even though GFP-Nur77-LBD colocalized extensively with mitochondria (Fig. 6c, d). Unlike Myc-Nur77, transfected Myc-Nur77-LBD also failed to mediate the mitophagic effect of celastrol (Supplementary Fig. 6a, b). This is reminiscent of the role of p62 in Parkin-mediated mitophagy, in which it serves only to cluster damaged mitochondria but not their autophagy[18,19]. Collectively, these results showed that Nur77 ubiquitination-mediated p62 phase separation is insufficient to direct targeted cargo to autophagosome and that the structural integrity of Nur77 is required to confer p62 with full autophagy receptor activity.

Our finding that p62 is required for Nur77-dependent mitophagy is interesting as p62 serves to sequester dysfunctional mitochondria but not their connection to the autophagy machinery in Parkin-dependent mitophagy[18]. To confirm the mitophagic role of p62 in Nur77-dependent mitophagy, we examined the colocalization of Nur77- and p62-positive condensates with core ATG proteins essential for autophagosome formation (Supplementary Fig. 6c–f). Our results showed that ULK1 and FIP200 failed to colocalize with celastrol-induced p62 condensates formed with Nur77 or Nur77-mutants (Nur77-ΔIDR and Nur77-K536R). In contrast, WIPI2 and ATG16L1 exhibited extensive colocalization with p62 condensates formed with Nur77 but not Nur77 mutants (Nur77-ΔIDR and Nur77-K536R). These results demonstrated that Nur77/p62-mediated mitophagy is independent on the ULK1/FIP200 complex but requires early autophagy proteins WIPI2 and ATG16L1 for autophagosome formation.

When p62 functions as a full autophagy receptor for protein aggregates, ubiquitin-induced p62 phase separation generates high avidity interaction with LC3 on the inner surface of the autophagosomal membrane, which is responsible for tethering of targeted cargo to autolysosome for autophagy[40]. Thus, we examined whether the lack of connecting Nur77-LBD/p62 condensates to the autophagy machinery was due to their inability to interact with LC3. Co-IP experiments demonstrated that p62 interacted with LC3 in the presence of Nur77 but not Nur77-K536R when cells were treated with celastrol (Fig. 6e), revealing a high-affinity interaction of Nur77/p62 complex with LC3, which is dependent on Nur77 ubiquitination. By contrast, cotransfection of Nur77-LBD, which binds p62 in a ubiquitination-dependent manner, could not induce p62 interaction with LC3 (Fig. 6f). Thus, p62 condensates induced by Nur77 ubiquitination alone is insufficient to direct cargo mitochondria to the autophagosome.

**The IDR of Nur77 interacts with PB1 of p62.** Our observation that ubiquitin-induced p62 phase separation is insufficient to complete mitophagy prompted us to study the role of Nur77-IDR in modulating the cargo activity of p62. To this end, we found that Nur77-IDR could form condensates with p62 in the absence of celastrol, even though the resulting Nur77-IDR/p62 condensates were not found either at mitochondria or in the lysosome (Fig. 6a–d). We first asked whether Nur77-IDR is involved in celastrol-independent interaction between Nur77 and p62 observed (Fig. 5b). Mutational analysis of p62 (Fig. 7a) demonstrated that deleting TB or ZZ domain from p62 had no effect on celastrol-dependent and -independent interaction between Nur77 and p62 (Fig. 7b), excluding their involvement. Removing the UBA domain impaired celastrol-dependent but not -independent interaction, in agreement with its role in binding ubiquitinated Nur77 (Fig. 5b). When PB1 was deleted, however, both celastrol-dependent and -independent interactions were lost, revealing a critical role of PB1 in p62 high-affinity interaction with Nur77. The importance of UBA and PB1 was also illustrated by confocal microscopy analysis showing that celastrol-induced p62 colocalization with Nur77 was impaired by their deletion (Supplementary Fig. 7a). Mutational analysis of Nur77 demonstrated that DBD of Nur77 (Nur77-ΔDBD) is dispensable for celastrol-dependent and -independent interactions (Supplementary Fig. 7b). When GFP-Nur77-IDR was analyzed, we found that it strongly interacted with Flag-p62 but not Flag-p62 lacking PB1 (Flag-p62-ΔPB1) in a celastrol-independent manner (Fig. 7c). It also interacted with p62-PB1 but not p62-UBA (Fig. 7d). For comparison, Nur77-LBD interacted with p62-UBA but not p62-PB1 in a celastrol-dependent manner (Supplementary Fig. 7c). Taken together, these data demonstrated that celastrol-induced formation of Nur77-p62 condensates is mediated by their unique celastrol-dependent and -independent interactions involving their

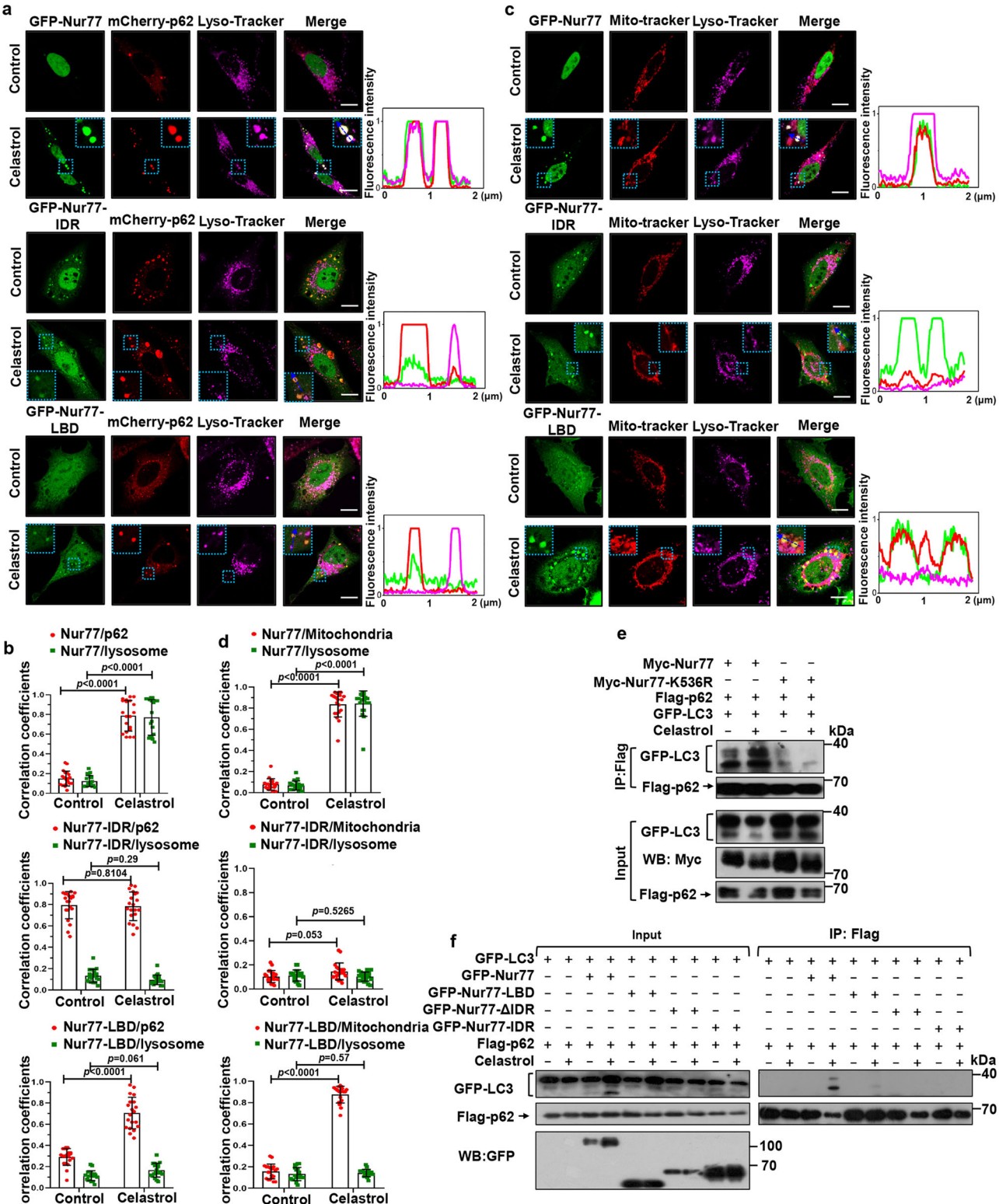

**Fig. 6 Nur77-LBD is insufficient to mediate celastrol-induced mitophagy. a–d** Representative images showing colocalization of Nur77 or mutants with p62, mitochondria, and lysosome in HeLa cells after celastrol treatment. The blue arrow indicates line profiles of fluorescence intensities including Pearson's correlation coefficients shown in **b** and **d**. A two-tailed unpaired Student's *t*-test was used for statistical analysis, and data were presented as mean values ± SEM (*n* = 20 biologically independent samples). Dotted box: higher magnification of indicated region. Scale bar, 10 μm. **e** Mutating K536 in Nur77 inhibits celastrol-induced interaction between p62 and LC3. HeLa cells transfected with the indicated expression plasmids were treated with or without 2 μM celastrol and 20 ng/mL TNFα. Interaction of Flag-p62 with GFP-LC3 was examined by co-IP assay. **f** Characterization of domain requirement of Nur77 for promoting celastrol-induced p62 interaction with LC3. HeLa cells transfected with the indicated Flag-p62, GFP-LC3, and GFP-Nur77 or mutant were treated with celastrol and TNFα for 1 h and analyzed for Flag-p62 interaction with GFP-LC3 by co-IP assay. Data represent at least three independent experiments. Source data are provided as Source Data file.

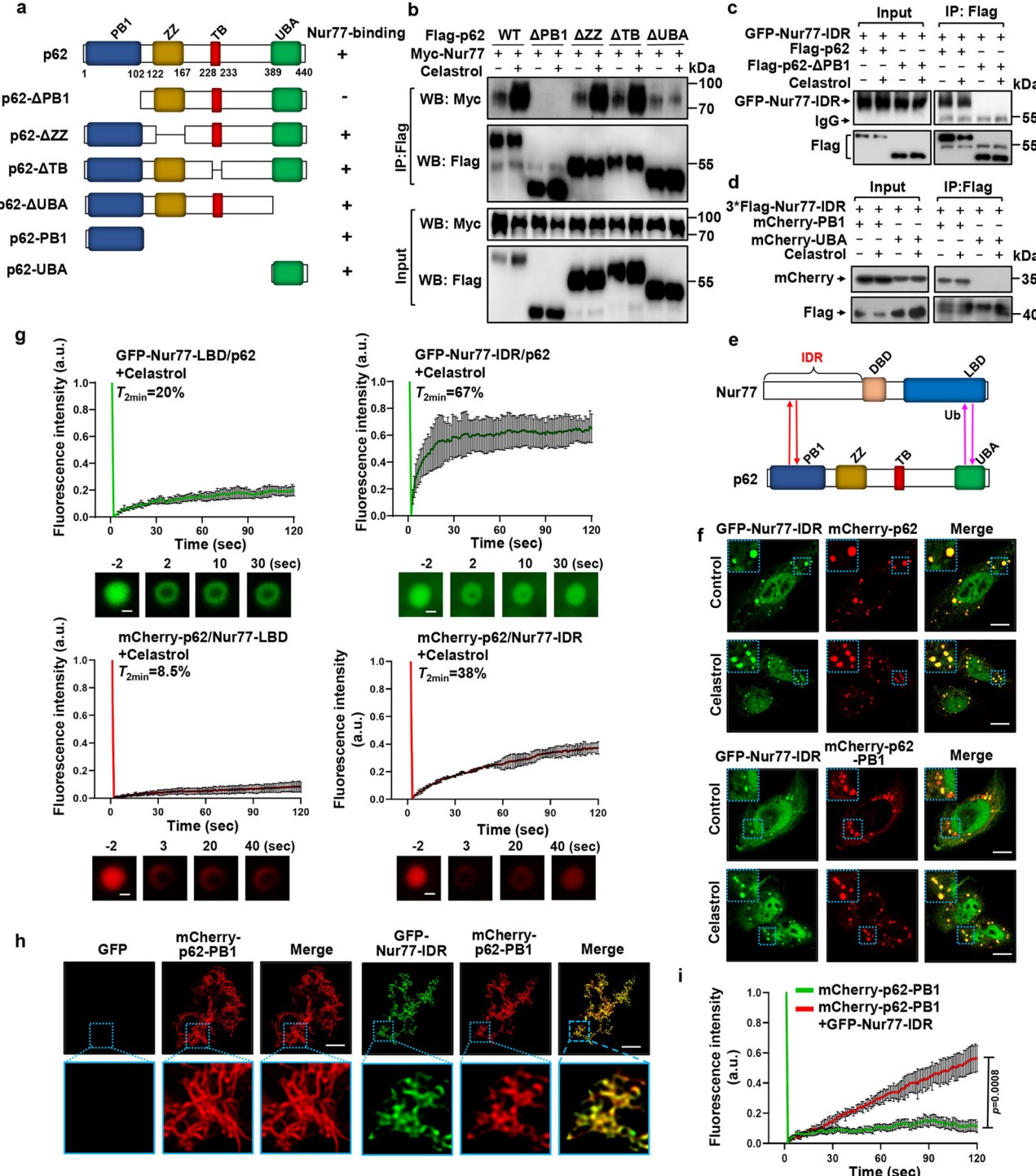

**Fig. 7 IDR interaction with PB1 promotes liquidity of p62 condensates. a** Schematic representation of p62 and mutants and their interaction with Nur77. PB1 Phox/Bem1p protein-protein binding domain. ZZ zinc-finger domain, TB TRAF6 binding domain, UBA ubiquitin-associated domain. **b–d** Interaction of Nur77 and p62, as well as their mutants, was analyzed in HeLa cells treated with or without celastrol by co-IP assay. **e** Multivalent interaction between Nur77 and p62. The interaction between the IDR of Nur77 and PB1 of P62 is ligand-independent (red), whereas the interaction between LBD of Nur77 and UBA of p62 depends on celastrol that triggers Nur77-LBD ubiquitination (pink). **f** Immunofluorescence images showing colocalization of GFP-Nur77-IDR with mCherry-p62 or mCherry-p62-PB1 after treatment with or without celastrol. Scale bar, 10 μm. **g** FRAP analysis of the effect of Nur77-IDR in regulating p62 mobility in HeLa cells. Data were presented as mean values ± SEM (*n* = 3 independent experiments). Scale bar, 1.5 μm. **h** Representative images showing the effect of GFP-Nur77-IDR on the filamentous structures of mCherry-p62-PB1 when mCherry-p62-PB1 was incubated with GFP or GFP-Nur77-IDR at intermediate molar ratio (1:2). Scale bar, 10 μm. **i** FRAP analysis of the effect of GFP-Nur77-IDR on mCherry-p62-PB1 mobility in vitro. Two-tailed unpaired Student's *t*-test was used for statistical analysis, and data were presented as mean values ± SEM (*n* = 3 independent experiments). Data represent at least three independent experiments. Source data are provided as Source Data file.

C-terminal and N-terminal regions in a "head-to-head" and "tail-to-tail" manner (Fig. 7e).

**Role of IDR of Nur77 in celastrol-induced mitophagy.** PB1 is known to mediate p62 oligomerization[15,58], which is critical for not only sequestering ubiquitinated cargo but also their connection to the autophagosome[14,15]. Given the above observation that Nur77-IDR could directly interact with p62-PB1, we suspected that the interaction might play a crucial role in celastrol-induced mitophagy. Nur77-IDR interacted with p62, resulting in the formation of Nur77-IDR/p62 condensates (Fig. 7f), which however were not found either on mitochondria (Fig. 5f) or in the lysosome (Fig. 6a). This suggested that the interaction is not directly involved in sequestering and tethering dysfunctional mitochondria, raising an intriguing question that the interaction might play a modulatory role in directing engulfed mitochondria to the autophagy machinery. Thus, we studied how IDR-dependent phase separations could confer Nur77-p62 condensates with the ability to connect targeted mitochondria to the autophagy machinery. We found that Nur77-IDR contributed significantly to the increase in the sizes and number of p62 body (Supplementary Fig. 7d), features that are important for effective autophagy[59]. The flexibility in the size of condensates allows the sequestration of different sizes and amounts of cargo for autophagy. In case of mitophagy, removing large organelle such as mitochondrion may depend on the size of the sequestering compartment, which is essential to maintain high-avidity interaction of p62 with ubiquitin and LC3. The liquidity of protein condensates is another critical determinant for their selective membrane sequestration, and a receptor with floatability is essential for selective autophagy[47]. Our FRAP analysis also revealed that the liquidity of p62 was largely dependent on Nur77-IDR but not Nur77-LBD, as Nur77-IDR increased significantly the mobility of p62 condensates with ~38% fluorescence recovery within 120 s (Fig. 7g). For comparison, Nur77-LBD and p62 showed ~8.5% fluorescence recovery, reflecting their lower fluidity. Incubation of purified mCherry-p62-PB1 with GFP-Nur77-IDR but not control GFP protein resulted in a change of mCherry-p62-PB1 from elongated filamentous to puncta structure (Fig. 7h), accompanied with increased liquidity (Fig. 7i). Taken together, our data demonstrated that Nur77 through its IDR drives mitophagy by increasing the size and fluidity of p62 condensates.

## Discussion

Phase separation plays a critical role in the condensation of misfolded and ubiquitin-positive proteins and their degradation by autophagy[37,38,60]. We present evidence here that coordinated phase separations of Nur77 and p62 through their unique "head-to-head" and "tail-to-tail" interactions assemble membraneless Nur77-p62 condensates capable of driving celastrol-induced mitophagy. During the celastrol-induced formation of Nur77-p62 condensates, the ubiquitination-dependent "tail-to-tail" interaction of Nur77 with p62 promotes p62 phase separation, which serves to prime and sequester dysfunctional mitochondria, whereas the "head-to-head" interaction involving IDR of Nur77 and PB1 of p62 confers the Nur77-p62 condensates with liquidity, which is necessary to connect targeted cargo mitochondria to the autophagic machinery (Fig. 8). These results reveal a spatio-temporal control of Nur77/p62 phase separation for both clustering dysfunctional mitochondria and tethering them to autolysosome for degradation.

Our data demonstrated that the sequestration of dysfunctional mitochondria by p62 is analogous to its clustering of ubiquitinated proteins, both of which require the formation of p62-

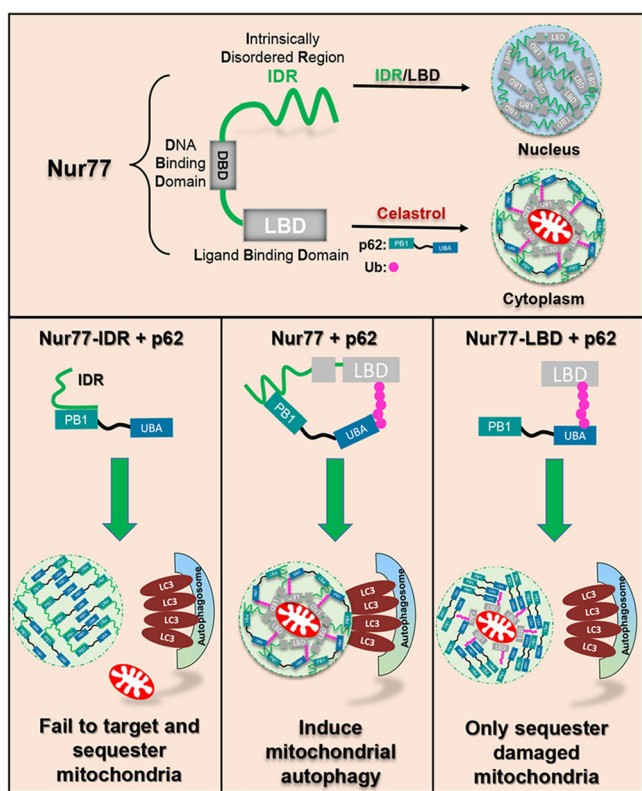

**Fig. 8 Graphic summary of Nur77 phase separation and its role in celastrol-induced mitophagy by promoting the liquidity of p62 condensates.** The phase separation of Nur77 and p62/SQSTM1 triggered by their multivalent interaction sequesters damaged mitochondria and directs cargo mitochondria to the autophagic machinery. DBD DNA-binding domain, LBD ligand-binding domain, IDR intrinsically disordered region, PB1 Phor and Bem1p, UBA ubiquitin-associating, Ub ubiquitin.

ubiquitin condensates. Although ubiquitin-induced p62 phase separation is sufficient to condensate ubiquitinated proteins and connect them to autophagosome[16,40], it is insufficient to direct clustered mitochondria to the autophagic machinery. This is illustrated by our data that the phase separation of p62 by ubiquitinated Nur77-LBD could sequester damaged mitochondria but failed to connect cargo mitochondria to autolysosome (Fig. 6a, c). The mechanistic basis for such a difference appears to rely on the ability of phase separated p62 condensates to interact with LC3, which is essential for connecting cargo to the autophagosome membrane[12]. p62–ubiquitin condensates sequestering misfolded proteins are able to interact with LC3 when engaged in phase separation[16,40], while p62–ubiquitin condensates clustering damaged mitochondria fail to connect cargo mitochondria to the autophagosome membrane (Fig. 6e, f). Thus, ubiquitin-induced p62 bodies is insufficient to direct cargo mitochondria to the autophagic machinery, reminiscent of the role of p62 in PINK1/Parkin-mediated mitophagy[18,19]. Selective autophagy in general requires a high density of receptors on the cargo surface in order to form tight and extensive contacts between the cargo and the autophagic membrane[13]. p62 employs oligomerization via its PB1 domain to generate high avidity interaction with LC3 and ubiquitin tags[14]. As LIR activity is subject to regulation by p62 oligomerization[13], it is possible that p62 oligomerization required to sequester ubiquitinated mitochondria may allosterically impair its LIR activity. Indeed, we showed that an additional interaction mediated by p62-PB1 and Nur77-IDR is required for efficient delivery of cargo mitochondria to autolysosome. Zaffagnini et al. also found that NBR1 interaction with the PB1 domain has a

direct stimulatory effect on substrate condensation and autophagy through its promotion of p62-mediated phase separation[40].

Recently, IDRs, especially those containing a low-complexity domain, have been shown to drive the formation of intracellular membraneless organelles for compartmentalized biochemical reactions by LLPS[61,62]. The N-terminal IDR of Nur77 exhibits liquid-like property, which is necessary not only for phase separation of Nur77 but also for promoting the expansion and migration of Nur77/p62 condensates. In the case of PGL granules in *C. elegans* embryos, cargo mobility is associated with increased PGL granule degradation by selective autophagy[48]. The liquidity of Ape1 droplets further strengthens the concept that the fluidity of biomolecular condensates determines their susceptibility to selective autophagy[47]. It is likely that there is optimal liquidity for biomolecular condensates to be an ideal cargo for selective autophagy of mitochondria. The p62 body, as a specific receptor protein, has been reported to be liquid-like in its droplet state, but with low mobility[16,40]. The behavior of condensates may depend on the physicochemical properties of the ubiquitinated substrates, and damaged mitochondria are likely to show less mobility than misfolded proteins. Thus, the liquid-like properties of Nur77-IDR could confer cargo mitochondria with fluidity for the high-avidity interaction of p62 with ATG proteins anchored on the autophagosomal membrane. How substrate sequestration and autophagosome formation are coordinated during mitophagy remains poorly understood. p62 has been implicated in the formation of autophagosome by driving the bending of the membrane around cargo[13,63]. The liquidity of p62 may also afford cargo with flexibility for tight and extensive contact with the nascent autophagosomal membrane through LC3 binding, a mechanism that was recently proposed to utilize cargo as a template for membrane formation[47]. Interestingly, Nur77/p62 bodies could colocalize with early autophagy proteins WIPI2 and ATG16L1 in an IDR-dependent manner (Supplementary Fig. 6e, f), raising the possibility that the fluidity of Nur77/p62 bodies may have a role in the initiation of autophagosome formation. Although inconclusive, we found that the interaction of Nur77-IDR with p62-PB1 could shorten p62-PB1 filaments (Fig. 7h), reminiscent of the binding of the autophagy receptor NBR1 to the PB1 of p62, which leads to shortening and solubilization of filamentous p62 structure[14]. The fact that overexpression of NBR1 can block the autophagic turnover of p62 bodies[64] suggests a potential regulatory role of NBR1 in Nur77-mediated mitophagy, which remains to be investigated.

Nur77 shares similar structural domains with other members of the nuclear receptor superfamily. Compared to LBD and DBD, the N-terminal IDR region is most variable among nuclear receptor family members but accounts for a large portion of nuclear receptor activities in a cell- and context-dependent manner[65]. Moreover, numerous nuclear receptor subtypes differing in their N-terminal A/B region are generated through alternative splicing and translation initiation[66] as well as proteolytic cleavage[54], providing the complexity of nuclear receptor regulatory networks and cell/tissue-specific signaling. As LLPS-mediated formation of membraneless organelles have emerged as important platforms to mediate diverse biological processes, our finding that the IDRs of Nur77 and its subfamily members could act by phase separation suggests that IDR-mediated phase separation may represent an important mechanism utilized by nuclear receptor family members to regulate diverse biological activities.

Membraneless condensates resulting from the interaction between Nur77 and p62 may also serve as an important signaling hub for diverse cellular events that two proteins are known to play, such as cell survival, apoptosis, metabolism, and amino acid sensing and the oxidative stress response[67,68]. Mitophagy is important for mitochondrial quality control, thereby contributing to cellular homeostasis and potentially preventing aggregation-induced diseases such as metabolic[69] or neurodegenerative diseases[70] and cancer[71]. The Nur77/p62-mediated mitophagy likely plays a crucial role in physiological and pathological conditions such as aging and inflammatory disease as well as cancer[2,72]. Our discovery that the IDR of Nur77 could promote the autophagy of aggregated mitochondria in a ligand-dependent manner offers an opportunity for developing new therapeutics for treating aggregation-associated diseases and cancer.

## Methods

**Plasmids and reagents**. Plasmids GFP-Nur77, GFP-Nur77-IDR, GFP-Nur77-ΔIDR, GFP-Nur77-LBD, GFP-Nur77-ΔDBD, GFP-Nur77-K536R, GFP-Nur77-LBD-K536R, mCherry-LC3, GFP-LC3, mCherry-p62, mCherry-p62-ΔPB1, mCherry-p62-PB1, mCherry-p62-UBA, mCherry-TRAF2, Myc-Nur77-ΔDBD, Myc-Nur77, Myc-Nur77-K536R, 3*Flag-Nur77-LBD, Flag-Nur77-LBD-K536R, Myc-p62, Flag-p62, Flag-p62-ΔPB1, Flag-p62-ΔTB, Flag-p62-ΔZZ, Flag-p62-ΔUBA, and HA-Ub were described previously[28]. Plasmid EGFP-mCherry-COX8 was from Addgene (Plasmid #78520)[42], and AAV-9-EGFP-mCherry-COX8 plasmids was subcloned by Genomeditech. Plasmids His-GFP-Nur77, His-GFP-Nur77-IDR, His-GFP-Nur77-LBD, His-mCherry-p62-PB1, His-mCherry-Nur77, His-mCherry-Nur77-IDR, and His-mCherry-Nur77-LBD, were generated with the pET-21a-mCherry or pET-28a-GFP vector using Hind III and Xho I restriction sites, respectively (Thermo Fisher Scientific). Nur77 (NM_002135, siRNA ID: SASI_Hs02_00333289, SASI_Hs02_00333290, and SASI_Hs02_00333291), SQSTM1/p62 (NM_003900, siRNA ID: SASI_Hs01_00118616, SASI_Hs01_00118618, and SASI_Hs01_00118620), Atg7 (NM_006395, siRNA ID: SASI_Hs01_00077648, SASI_Hs01_00077649, and SASI_Hs02_00341471) and control siRNAs from Sigma-Aldrich. siRNA was transfected into cells using Lipofectamine 2000 transfection reagent (Thermo Fisher Scientific, 11668019) according to the manufacturer's protocols. Control siRNA served as a negative control.

**Antibodies**. For Western blot, the following antibodies were used: Myc (9E10) (Santa Cruz, sc-40, dilution: 1:1000), FLAG (Sigma-Aldrich, F1804, dilution: 1:1000), GFP (Santa Cruz, sc-9996, dilution: 1:1000), COXII (Abclonal, A3843, dilution: 1:1000), LC3 (Abcam, AB51520, dilution: 1:5000), Nur77 for HeLa cell study (CST, 3960S, dilution: 1:500), Nur77 for mouse study (Thermo Fisher Scientific, 14-5965-82, dilution: 1:1000), SQSTM1/p62 (Abcam, AB56416, dilution: 1:5000), Atg7 (HUABIO, SC06-30, dilution: 1:1000), β-actin (Sigma-Aldrich, A5441, dilution: 1:10,000), GAPDH (Proteintech, 60004, dilution: 1:10,000). The goat anti-mouse IgG F(ab′)2 secondary antibody (#31436, dilution: 1:10,000) and the goat anti-rabbit IgG F(ab′)2 secondary antibody (#31461, dilution: 1:10,000) were from Pierce Chemical. For immunoprecipitation, FLAG (Sigma-Aldrich, F1804, dilution: 1:100) was used. For immunostaining, the following antibodies were used: Nur77 for HeLa cell study (CST, 3960S, dilution: 1:100), Nur77 (M-210) (Santa Cruz, sc-5569, dilution: 1:50), Nur77 for mouse study (Affinity, DF7850, dilution:1:200), CD68 (Abcam, AB955, dilution: 1:200), SQSTM1/p62 (Abcam, AB56416, dilution: 1:400), Myc (9E10) (Santa Cruz, sc-40, dilution: 1:100), FLAG (Sigma-Aldrich, F1804, dilution: 1:400), HA (Santa Cruz, sc-7392, dilution: 1:100), LC3 (MBL, PM036, dilution: 1:200), Hsp60 (Santa Cruz, sc-13115, dilution: 1:100), FIP200 (CST, 12436, dilution: 1:100), ATG16L1 (Proteintech, 19812, dilution: 1:100), WIPI2 (Proteintech, 15432, dilution: 1:100), ULK1 (Santa Cruz, Sc-390904, dilution: 1:50), MitoTracker Red FM (Thermo Fisher Scientific, M22425, dilution: 1:20000), MitoTracker Deep Red FM (Thermo Fisher Scientific, M22426, dilution: 1:20000), LysoTracker Deep Red FM (Thermo Fisher Scientific, L12492, dilution: 1:20,000), Goat anti-Rabbit (A10523) and anti-Mouse (A10524) IgG (H+L) Cross-Adsorbed Secondary Antibody, Cy5 (Thermo Fisher Scientific, dilution: 1:200), Cy3-AffiniPure Goat Anti-Rabbit (111-165-003) and Anti-Mouse (115-165-003) IgG (H+L) (Jackson, dilution: 1:200), FITC-AffiniPure Rabbit Anti-Goat IgG (H+L) FITC (Yeasen, 33707ES60, dilution: 1:200). For Immunogold EM, the following antibodies were used: Nur77 (CST, 3960S, dilution: 1:100), SQSTM1/p62 (Abcam, AB56416, dilution: 1:100), LC3 (MBL, PM036, dilution: 1:200), Goat Anti-rabbit IgG/Gold (AB-0295G-Gold, 15 nm), Goat Anti-mouse IgG/Gold (AB-0296R-Gold, 10 nm), (Leading Biology Inc., California, USA), dilution: 1:100).

**Cell culture**. Human cervical cancer cell line HeLa and *Nur77*[−/−]HeLa[53] were cultured in RPMI-1640 (Sigma-Aldrich) with 10% fetal bovine serum (FBS), 100 U/ml penicillin-streptomycin (Sigma-Aldrich). HEK293T cells were maintained in DMEM (Sigma-Aldrich) containing 10% FBS and grown under standard tissue-culture conditions (37 °C, 5.0% $CO_2$). Primary mouse embryonic fibroblasts (MEFs) and *p62*[−/−]MEFs were obtained from M.T.D.-M and J.M.'s lab[73]. Transfection was performed using lipofectamine 2000 or PEI reagent according to the manufacturer's instructions (Thermo Fisher Scientific).

**CRISPR genome editing**. Nur77 knockout cells were generated using CRISPR-Cas9 methods. Briefly, the design of single-guide RNAs (sgRNAs) was based on recommendations from the Zhang laboratory website (http://crispr.mit.edu/). gRNA targeting sequence of Nur77 (5′-ACCTTCATGGACGGCTACAC-3′) was cloned into gRNA cloning vector Px330 (Addgene, 71707) and confirmed by sequencing. To screen for cells lacking Nur77, HeLa cells were transfected with control vector and gRNA expression vectors, followed by G418 selection (0.5 mg/ml). Single colonies were subjected to Western blotting using anti-Nur77 antibody to select knockout cells, which were validated by STR typing (Genetic testing biotechnology, Suzhou, China).

**Mice**. Wild-type (C57BL/6 J, Stock No.: 000664) and *Nur77*$^{-/-}$ mice (Nur77 C57BL/6 J, Stock No: 006187) were purchased from the Jackson Laboratory (Bar Harbor, Maine, USA). All experiments were performed on female cohorts at 2 months of age (young mice) and 2 years of age (aging mice). All mice were housed in pathogen-free facilities, in a 12 h dark/light cycle with temperatures of 22.5 °C and 50–55% humidity. The protocols for animal studies were approved by the Animal Care and Use Committee of Xiamen University, and all mice were handled in accordance with the "Guide for the Care and Use of Laboratory Animals" and the "Principles for the Utilization and Care of Vertebrate Animals".

**Protein expression and purification**. Human Nur77 and mutants were cloned into a modified version of a T7-promoter-pET expression vector. The base vector was engineered to include a 6×His tag followed by either GFP or mCherry. The resulting expression constructs were confirmed by sequencing. The N-terminal 6×His tagged GFP, GFP-Nur77, GFP-Nur77-IDR, and GFP-Nur77-LBD and C-terminal 6×His tagged mCherry, mCherry-Nur77, mCherry-Nur77-IDR, and mCherry-Nur77-LBD were overexpressed in *Escherichia coli* BL21 (DE3) cells (Novagen) using pET-28a-GFP or pET-21a-mCherry vectors. Proteins were induced with 0.5 mM isopropyl β-d-1-thiogalactopyranoside (IPTG) for 16 h at 16 °C. Cells were collected, pelleted, and then resuspended in the following buffer: 50 mM Tris-HCl pH 7.4, 400 mM NaCl, 20 mM imidazole, supplemented with EDTA-free protease inhibitor cocktail (Roche) according to the manufacturer's instructions. Cells were then lysed by sonication and centrifuged at 7000$g$ for 30 min. The supernatants were first purified with Ni-NTA resin (GenScript), followed by purification on a Superdex 200 increase 10/300 column (GE healthcare). All proteins were stored in storage buffer (50 mM Tris-HCl pH 7.4, 150 mM NaCl) at −80 °C. All protein purification steps were performed at 4 °C. Purified protein was quantified using a ND-2000C NanoDrop spectrophotometer (NanoDrop Technologies) with OD 280 and verified by Coomassie staining.

**Co-immunoprecipitation (Co-IP) and Western blot (WB)**. Cells transfected with indicated expression vectors were treated with celastrol (2 μM) for 1 h and analyzed by Co-IP using anti-Flag antibody. Cells were harvested in lysis buffer (25 mM Tris-HCl pH 7.4, 150 mM NaCl, 1% Triton X-100, 5 mM ethylenediaminetetraacetic acid, and protease inhibitors). The lysate was incubated with 1 mg antibody at 4 °C for 2 h. Immunocomplexes were then precipitated with 30 μl of protein A/G-Sepharose. After extensive washing with lysis buffer, the beads were boiled in sodium dodecyl sulfate (SDS) sample loading buffer, analyzed by 10% SDS–polyacrylamide gel electrophoresis (SDS–PAGE), and transferred to nitrocellulose. The membranes were blocked in 5% milk in Tris-buffered saline and Tween 20 (TBST; 10 mM Tris–HCl pH 8.0, 150 mM NaCl, and 0.05% Tween 20) for 1 h at room temperature. After washing twice with TBST, the membranes were incubated with appropriate primary antibodies in TBST for 1 h, washed twice, and then probed with horseradish peroxide-linked anti-immunoglobulin (1:5000 dilution) for 1 h at room temperature. After three washes with TBST, immunoreactive products were visualized using enhanced chemiluminescence reagents and autoradiography.

**Immunofluorescence**. Cells were grown in 24-well plates with 0.5 mm × 0.5 mm glass slides and then permeabilized with PBS containing 0.1% Triton X-100 and 0.1 mol/L glycine for 15 min, and blocked with 1% bovine serum in PBS for 30 min at room temperature, followed with incubation with various primary antibodies at room temperature for 3 h, and detected by FITC-labeled anti-IgG, anti-goat IgG conjugated with Cy3, or Cy5-labeled antibody at room temperature for 1 h. Cells were costained with 40,6-diamidino-2-phenylindole (DAPI) (1:10,000 dilution) to visualize nuclei. The images were taken under a fluorescent microscope Zeiss LSM 880 (Zeiss, Germany) or Leica TCS SP8 (Leica Microsystems GmbH, Mannheim, Germany) confocal laser scanning microscope system. HeLa, Nur77$^{-/-}$HeLa, MEFs, and p62$^{-/-}$MEFs were treated with 2 μM celastrol or plus TNFα (20 ng/mL) for the indicated time. Mitochondria and lysosome were marked by Mito-tracker and lyso-tracker for 30 min before being fixed by 4% buffered formalin/PBS.

**Immunostaining on cryosections**. Livers were dissected, cut into small pieces, and fixed in 4% paraformaldehyde at 4 °C overnight. After washes, sink the samples in 10% sucrose solution for 4 h, 15% sucrose solution for 4 h, and 20% sucrose solution overnight at 4 °C. The next day, samples were embedded in OCT (optimum cutting temperature compound). Samples were sliced into 4-μm sections and incubated with anti-LC3B antibody, anti-Nur77, anti-Hsp60, anti-p62, or anti-CD68 antibody. In the in vivo mitophagy assay, $2 \times 10^{11}$ VG/ml of AAV-9-EGFP-mCherry-COX8 plasmids were injected into young and old mice by intravenous injection. Two weeks after injection, the liver tissues were fixed and frozen with the above steps. Sections were mounted on slide glasses. The section slips were examined and analyzed by the Leica TCS SP8 system.

**Immunogold EM**. HeLa cells were transfected with Myc-Nur77 and Flag-p62 plasmids for 24 h, and then treated with 2 μM celastrol and 20 ng/ml TNF-α for 0.5 h. HeLa cells were fixed with paraformaldehyde and 0.1% glutaraldehyde in 0.1 M sodium phosphate buffer (pH 7.4) in 4 °C overnight. After dehydration using 30, 50, 70, 90, and 100% ethanol and infiltration in LR resin. Ultrathin sections of 90–100 nm were obtained. Sections were blocked with 1% BSA in PBS for 10 min in room temperature, and then incubated with monoclonal anti-p62 and anti-Nur77, or anti-p62 and anti-LC3 in 4 °C overnight. Cells were washed three times with ddH$_2$O, followed incubation with 10 nm gold-conjugated goat antibodies to mouse IgG and 15 nm gold-conjugated goat antibodies to rabbit IgG for 3 h at room temperature. After washing five times with ddH$_2$O, cells were further fixed with 2.5% glutaraldehyde and washed three times with ddH$_2$O. Immunogold-labeled samples were stained with uranium and lead citrate before imaging with a Hitachi HT-7800 electron microscope.

**Imaging and quantification of mitophagy**. All images were acquired with Leica TCS SP8 (Leica) confocal microscope with a Plan-Apochromat 63x oil objective. The 488 and 561 nm laser lines were used to excite EGFP-mCherry-COX8 and imaging was done in line mode to minimize movement of mitochondria between the acquisition of each channel. The 633 nm laser line was used to excite Alexa 633-conjugated dyes. Cells were transiently transfected with EGFP-mCherry-COX8 plasmid to examine the formation of fluorescent puncta of autophagosomes. After transfection, the cells were treated with DMSO or 2 μM celastrol and 20 ng/ml TNFα for 3 h, and then examined by immunostaining. For rescue experiments, *Nur77*$^{-/-}$HeLa cells were cotransfected with Nur77 or mutants together with EGFP-mCherry-COX8 and then treated with celastrol and TNFα. Quantitative analysis of mitophagy was assessed by the cytoFLEX Flow Cytometry System (Beckman-Coulter, Miami, FL, USA). Flow cytometry scatter illustrate the gating strategy (Supplementary Fig. 1a) used to detect the mCherry-COX8 exposure cells. EGFP-mCherry-COX8 positive cells were first gated (gate: P1) based on forward scatter area (FSC-A) and side scatter area (SSC-A), followed by doublet discrimination based on FSC height (FSC-H) by FSC-A (gate: Single Cells). EGFP-mCherry-COX8 positive cells were determined using untransfected controls (gate: FITC). Measurements EGFP-mCherry-COX8 were made using dual-excitation ratiometric pH measurements at FITC (488 nm, pH 7) and PE (561 nm, pH 4) lasers. For each sample, 10,000 events of FITC positive cells were collected and subsequently gated with appropriate controls to detect mCherry-COX8 exposure cells but not EGFP-mCherry-COX8 double-positive cells. Three independent experiments were conducted and one of three similar experiments is shown.

**Live cell imaging**. Live cell imaging was carried out in a normal culture medium. For HeLa cells, imaging was performed using Zeiss 880 LSM with Airyscan, plan-apochromatic 63x oil objective, and processed with Airyscan processing in ZEN software (Zeiss). GFP-Nur77 trafficking imaging was performed using a Nikon spinning disk equipped with a 63x oil objective lens. Time-lapse movies of GFP-Nur77 and mCherry-p62 were then acquired at 100 ms/frame for 30 s and displayed in kymograph format.

**Intrinsic disorder tendency analysis**. Intrinsic disorder tendency of Nur77, NOR1, and Nurr1 was analyzed using Predictor of Natural Disordered Regions (PONDR) (http://www.pondr.com/) version SVL2 across the entire Nur77. The scores were assigned between 0 and 1, and a score above 0.5 indicates disorder.

**Fluorescence recovery after photobleaching (FRAP) analysis**. For in vitro experiments, FRAP was carried out with samples in 96-well microscopy plates using Zeiss LSM 880 microscope equipped with 63× oil immersion objectives. Droplets were bleached with a 488-nm laser pulse (three repeats, 80% intensity, dwell time 1 s). The recovery from photobleaching was recorded for the indicated time. For in vivo experiments, FRAP of GFP-Nur77, mCherry-p62, and their mutants was performed on the same fluorescent microscope mentioned above using 63× oil immersion objectives. Bleaching was performed using 80% laser power (488 or 561 nm laser) with 120/180 s, and images were collected every 1 s. Fluorescence intensity at the bleached spot, a control unbleached spot, and background was measured using the FIJI plugin FRAP Profiler. Background intensity was subtracted, and values are reported relative to the unbleached spot to control for photobleaching during image acquisition. Each data point is representative of the mean and standard deviation of fluorescence intensities in three unbleached (control) or three bleached (experimental) granules.

**In vitro phase separation assay**. Recombinant GFP or mCherry fusion proteins were concentrated and desalted to an appropriate protein concentration which contain 50 mM Tris-HCl pH 7.4 and 150 mM NaCl by Amicon Ultra centrifugal

filters (10 K MWCO, Millipore). In vitro droplet assembly for GFP-Nur77, GFP-Nur77-IDR, GFP-Nur77-LBD, or mCherry-Nur77 was mixed with 10% (w/v) final concentration of PEG-3.35 K (Sigma-Aldrich, 25322-68-3) or other crowding reagents (Sigma-Aldrich). Co-phase separations between GFP-Nur77-IDR and mCherry-Nur77, mCherry-Nur77-IDR or mCherry-Nur77-LBD were done by mixing in the droplet-formation buffer: 50 mM Tris-HCl pH 7.4, 150 mM NaCl, 10%PEG-3.35 K. All droplets were loaded onto a glass slide, covered with a coverslip, and imaged with Leica TCS SP8 microscope equipped with 63× or 100× oil immersion objectives.

**Turbidity and size of droplet assembly**. The size of the droplet was determined using a Malvern zetasizer (Malvern instrument, Worcestershire, UK) apparatus. After measuring the turbidity of the protein phase-separated solution by absorbance at 600 nm using BioTek Instruments (Winooski, VT, USA). All experiments were performed at least three times with similar results.

**Statistical analysis and reproducibility**. All data were presented as the mean values ± SEM of at least three technical replicates. Two-tailed unpaired Student's t-test were used for statistical analysis using the GraphPad Prism 8.0 software. Significance was defined as any statistical outcome that resulted in a $P$ value of < 0.05 unless otherwise indicated. Each experiment was independently repeated at least three times with similar results.

**Reporting Summary**. Further information on research design is available in the Nature Research Reporting Summary linked to this article.

## Data availability

All other relevant data supporting the key findings of this study are available within the article and its Supplementary Information files or from the corresponding author upon reasonable request. Source data are provided with this paper.

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

## Acknowledgements

We thank M.T.D.-M and J.M. for providing primary MEFs and p62$^{-/-}$MEFs[73]. This work was partially supported by grants from the Natural Science Foundation of China (81672749, U1405229, 81741171, and 91429306), Regional Demonstration of Marine Economy Innovative Development Project (16PYY007SF17), Fujian Provincial Science & Technology Department (2017YZ0002), Xiamen Bureau of Science & Technology (3502Z20193004 and 3502Z20150007), and the Postdoctoral Science Foundation of China (2020M671945).

## Author contributions

S.-z.P. and X.-h.C. conducted cellular and molecular experiments; S.-j.C., J.Z., D.Z., W.-r.L., and C.-y.W. provide technical assistance; S.-z.P., X.-h.C., Y.S., and X.-k.Z. analyzed data; S.-z.P., X.-h.C., Y.S., and X.-k.Z. wrote the manuscript.

## Competing interests

The authors declare no competing interests.

## Additional information

**Peer review information** *Nature Communications* thanks Masaaki Komatsu, Wen-Xing Ding and the other anonymous reviewer(s) for their contribution to the peer review this work. Peer reviewer reports are available.

