## [Peer Review File · Nature Communications]

Reviewers' Comments:

Reviewer #1:

Remarks to the Author:

In this manuscript, Peng et al., demonstrate that phase-separated Nur77 increases fluidity and size of p62 in a its ubiquitination dependent manner to induce celastrol-dependent mitophagy. The study is well-conducted, and the manuscript is written clearly. Liquid-liquid phase separation (LLPS) in selective autophagy is attracting interest in autophagy-research field, and this study provides a novel insight of LLPS into regulation of the mitophagy. Major concerns are lacking a presentation of physiological significance of Nur77- and p62-mediated mitophagy and analyses of autophagy-related proteins on Nur77- and p62-condensates.

Major comments

1. Throughout the manuscript, the authors utilized a triterpenoid antioxidant compound, celastrol to induce the mitophagy. What physiological conditions induces the Nur77- and p62-mediated mitophagy? Namely, what is a physiological role of Nur77- and p62-mediated mitophagy?
2. In the case of selective autophagy for Ape1 complex, P-body, and stress granule, liquidity or interacting protein(s) have been shown to be crucial factors for the structures to be surrounded by an autophagosome (Yamasaki A, et al. Liquidity Is a Critical Determinant for Selective Autophagy of Protein Condensates. *Mol Cell* 77, 1163-1175 e1169 (2020), Zhang G, Wang Z, Du Z, Zhang H. mTOR Regulates Phase Separation of PGL Granules to Modulate Their Autophagic Degradation. *Cell* 174, 1492-1506 e1422 (2018), Turakhiya A, et al. ZFAND1 Recruits p97 and the 26S Proteasome to Promote the Clearance of Arsenite-Induced Stress Granules. *Mol Cell* 70, 906-919 e907 (2018)). To prove whether the fluidity and size of Nur77- and p62-positive condensates determine autophagosome formation around mitochondria, the authors should investigate the localization of core ATG proteins essential for autophagosome formation such as FIP200, ULK1, WIPI2, ATG16L1 and LC3 on the condensates. And, also should verified failure of such localization in the case of Nur77-mutants including K567R and Δ IDR.
3. The formation of the autophagosome around phase-separated p62 requires sequential and antagonistic steps. First, a complex between p62 and FIP200 is formed to initiate the selective autophagy of p62. This complex is mutually exclusive to the one formed between p62 and LC3, indicating that the binding of p62 to LC3 occurs after dissociation from FIP200 (Turco E, et al. FIP200 Claw Domain Binding to p62 Promotes Autophagosome Formation at Ubiquitin Condensates. *Mol Cell* 74, 330-346 e311 (2019).) The authors should investigate whether Nur77 have an effect on the interaction.
4. To show autophagy-dependency, the authors should show the impairment of the celastrol-dependent mitochondria degradation in autophagy-deficient cells.
5. p62-condensates could be identified by their unique morphology by electron microscopy (EM) (Jakobi AJ, et al. Structural basis of p62/SQSTM1 helical filaments and their role in cellular cargo uptake. *Nat Commun* 11, 440 (2020).). The authors can show fine structure of Nur77- and p62-condensates around mitochondria by EM and compare them with p62-structures formed under stress condition. They may be also able to observe how isolation membranes sequester the mitochondrion.
6. In Figure 5b, the quantity of Nur77 to bind to p62 dramatically decreased by K536R mutation. Nevertheless, the K536R mutation did not affect co-localization efficacy with p62 (Fig. 5d).
7. In Figure 7a, distinct to Fig. 5b, the interaction of p62 with Nur77 in normal conditions was not detected. Nevertheless, the authors claimed that "Removing UBA domain impair celastrol-dependent but not independent interaction". The bands corresponding to Nur77 were broad and obscure.
8. The immunoblot in Figure 7b should be also replaced with representative results.
9. Extended Data Figure 1a: in the text (Figure legend), the authors described "TNF- α -treatment". But, in the Figure, the data is missing. Mislabeled?
10. Extended Data Figure 7a: immunoblot data should be replaced with representative one.
11. Recent study showed that overexpression of NBR1 blocks the autophagic turnover of p62-structures (Sanchez-Martin P, Sou YS, Kageyama S, Koike M, Waguri S, Komatsu M. NBR1-mediated p62-liquid droplets enhance the Keap1-Nrf2 system. *EMBO Rep* 21, e48902 (2020)). The quantity of NBR1 bound to p62-structures might regulate fluidity and size of p62 and in turn affect the direction of autophagic engulfment on the p62-condensates. The binding Nur77 to p62 might be competitive to NBR1-binding. The authors may discuss this point.

12. The pioneer studies showing the roles of p62 in selective autophagy should be cited. Bjorkoy G, et al. p62/SQSTM1 forms protein aggregates degraded by autophagy and has a protective effect on huntingtin-induced cell death. *J Cell Biol* 171, 603-614 (2005).
Komatsu M, et al. Homeostatic levels of p62 control cytoplasmic inclusion body formation in autophagy-deficient mice. *Cell* 131, 1149-1163 (2007).

Reviewer #2:

Remarks to the Author:

Peng et al., have presented a very interesting study following their previous work where they have discovered that mitochondrial Nur77 can mediate autophagy of dysfunctional mitochondria upon binding to celastrol. Here they demonstrate novel mechanistic insights on how Nur77 and p62 can coordinate (via phase separations) to sequester damaged mitochondria and to connect targeted cargo to the autophagic machinery. It is a very interesting and well-written manuscript. I do however have a couple of concerns that need to be addressed.

1) The authors should provide some more information on Celastrol and its use. They only mention that it is "a potent anti-inflammatory triterpenoid quinone-methide". They should provide more information, is it often clinically used? To treat which cases? If known: what are known effects and on which cells?

2) Figure 1A: the authors also should provide a zoom inset of the control cells.

Are the microscopy settings the same in control cells and stimulated cells? The fluorescence intensity of the control seems to be less.

Figure 1B and D: Next to the FACS plots, the authors should provide a quantification of at least 3 independent experiments and perform proper statistics to show variability and reproducibility of the experiment. In addition, they should provide example plots of the gating strategy (were the doublets and dead cells excluded?). The materials and methods section also should contain a paragraph describing the flow cytometry experiments.

3) Figure 2A and B: The pictures of these figures are stills of extended video 1. Data from multiple cells should be shown.

Quantification of the number and the size of the vesicles per cell would provide additional insight and will show the increase over time (including variability and reproducibility of the experiment). From these stills it is difficult to see what the increase over time is, especially since already more than 5 puncta are present at 0 min celastrol.

Important: The quantification should be not be done on overexposed pictures, as shown for the GFP channel in these figures.

(same is true for extended figure 1b, there is overexposure of GFP-nur77 and mCherry-p62, therefore it is not possible to correctly plot their fluorescence intensities)

Line 167: "the condensates of p62 and LC3 had a greater diameter compared with those of Nur77 and mitochondria" to make this statement, quantification of the diameter of multiple droplets per cell and multiple cells should be done incl proper statistics. Additionally a color swab needs to be performed to exclude diffraction effects. Additionally, quantification of diameter of vesicles should not be done on overexposed samples as shown in extended data fig2a and 2b

Figure 2C, why is the GFP channel in the control much less intense compared to the celastrol stimulated cells? To be able to properly do the quantification as the authors have shown in figure 2c in the Nur77^{-/-}-HeLA+GFP-Nur77 experiment, similarly exposed pictures should be compared.

4) Figure 3d, f, g, h reveals that Nur77 also forms puncta in the nucleus. Also Figure 2a, b and c (GFP-Nur77) should show this, but since these pictures are overexposed, this is not visible.

Therefore in the whole manuscript only pictures should be shown where samples are not overexposed.

5) Suppl figure 3c (30% PEG-3.35k) and 3d (5uM mCherry-Nur77); these picture show rather amorphous condensates than spherical droplets. Can the size of these condensates then properly be measured? Isn't it better to exclude these datapoints from the graphs?

6) The legend of figure 4 should include more details.

Are the cells stimulated with celastrol in fig 4c? For 4c the authors also should quantify the diffuse, diffuse/punctate and punctate cells for these mutants in HeLa cells as they did for the purified mutants in extended figure 4. The authors also should show the variability between independent experiments.

The DAPI channel is hardly visible and for 4c: the authors also state: "while deleting the DBD had little effect": why is this stated? This figure shows (n=1 cell) that there are no true spherical droplets.

4d: does nuclear export from the nucleus to the cytoplasm happen in all cells? Or in a proportion of the cells? Does this not happen in untreated cells? What is the difference? To be able to draw conclusions on this somehow this needs to be quantified.

4e: why was only an RXR ligand taken along, and not another Nur77 ligand, such as 6-MP, C-DIMS or cytosporone B? This could proof whether it is dependent on Celastrol or merely ligand binding to Nur77

7) Schematic drawings for the mutants used in figure 4 (Nur77 and P62) would be helpful.

8) Extended figure 7; what is compared with the statistics in figure D?

9) For clarity it would be better to include in all legends which statistical test was used, including the n. Not only in the materials and methods section.

10) In the discussion line 475 the authors state: "IDR-mediated phase separation may represent an important mechanism utilized by nuclear receptor family members to regulate diverse biological activities." This indeed may be true. Did the authors compare the IDR of Nur77 (NR4A1) with its family members Nurr1 (NR4A2) and NOR1 (NR4A3)? And did they ever investigate whether nurr1 and NOR1 also respond to celastrol? I think the discussion misses a paragraph discussing this.

In conclusion, the authors describe an interesting novel mechanism for phase separation of Nur77 important in mediating celastrol-induced mitophagy. However it is not clear whether this phenomenon is only present in this specific case (celastrol) or whether this could be a more general mechanism. Moreover, many figures only show pictures of a single cell, some conclusions are based on overexposed samples and statistics used is unclear. These are essential points that need to be addressed to proof the statements made.

Reviewer #3:

Remarks to the Author:

The work of Peng et al. investigates the phase separation of Nur77 mediated by celastrol and its involvement in mitophagy. The study demonstrates that Nur77 undergoes phase-separation when mixed with crowding agents in vitro, reproducing the experimental observation of liquid-like puncta formation inside the nucleus of cells. Most importantly, the study: i) dissects the interplay between Nur77 and p62 and their recruitment in phase separated compartments within the cell, ii) identifies the regions and modifications controlling their interaction (and consequently their phase-separation).

The observations are convincing, and the body of work is remarkable for scope and number of experimental conditions investigated. However, part of the data interpretation is problematic when framed in the context of phase separation and these aspects should be addressed to provide a consistent connection between the experimental observations and the underlying invoked physics. Finally, given the functional relevance and multi-layer structure of these membraneless compartments, I expect this publication can be of interest for a large audience. Therefore, I would suggest improving readability to help the non-specialized audience in the navigation of the large number of experimental conditions explored in the manuscript.

In details:

1. The manuscript would benefit of a scheme of the protein components and their architecture (as described in the introduction of the paper). This will ease the reader in remembering which domains of the proteins are involved in which interactions or modifications. This is done for Nur77 in Fig. 4 but not for p62. Similarly, the same scheme can be used across figures to identify which components are tested.

2. Similarly, a graphic summary of the emerging model would help understanding the different steps controlling cellular localization of the components, phase-separation, modification, cargo binding, and how these steps can lead to mitophagy.

3. Statement like: "The Nur77/p62 condensates require an additional phase separation mediated by the interaction of N-terminal intrinsically disordered region" (line 117-120) and similar ones in the discussion section (line 434-436) are highly confusing. Unless Nur77 undergoes proteolysis

and it is divided in two segments (IDR and LBD) within the cell, it is not simple to talk about “an additional phase separation”. The phase separation propensity (phase boundaries) and properties (e.g. fluidity) of fragments of the protein do not provide direct insights on the behavior of Nur77 phase separation propensity and properties, because for the full-length protein there will be not a distinct phase separation of the IDR or folded domain (giving rise to two phase separations), but the whole protein will participate into a single phase separation (even though multiple phases can coexist). There are particular cases in which a micro-phase separation can be invoked with specific domains of the protein partitioning in a specific phase (as for block co-polymers) but this is not supported by the presented data. Indeed, all the experiments point to the fact that both Nur77 and p62 are multivalent, with part of the multivalence aided by celastrol and ubiquitination. This is an important result that has to be stressed more.

4. At line 167, LC3 phase separation is mentioned in comparison to Nur77. Though previously introduced in the text, it was not obvious if the protein itself was undergoing phase separation together with p62. A sentence clarifying this point may ease the reader in following the experimental design. The corresponding outcome is also extremely important and parallels previous finding from other groups (e.g the Kriwacki lab on the organization of nucleophosmin inside of the nucleolus) on the formation of multi-shell condensates. A mention to previous literature in this respect would help interpreting the possible mechanisms leading to these types of condensates.

5. The authors observed that Nur77-IDR undergoes phase separation in cells, whereas Nur77-Delta-IDR, LBD, or Delta-DBD reduces or even suppress phase separation in absence of celastrol. Whereas it is clear that Nur77-IDR has an essential role in driving the phase-separation, observations that Nur77-LBD colocalizes with Nur77-IDR droplets suggests that heterotypic interactions between Nur77 domains may play a role. In this respect, it would be helpful to cast these experiments in the light of testing the role of homotypic and heterotypic interactions across domains of Nur77.

6. In the paragraph describing the celastrol regulation on Nur77 phase separation, at line 251 the authors state that celastrol promotes not only the nuclear export of Nur77 but also its phase separation. This is not yet clear from the data shown at this point of the paper. Celastrol promotes the nuclear export of the protein and phase-separation could be simply an outcome of the protein accumulation in the cytosol. Part of the subsequent experiments proves that celastrol can directly mediate the interaction between proteins, supporting the hypothesis that is also regulating phase-separation. Finally, from reading the manuscript it is unclear what K-80003 (in extended figure 4 it is referred to as K-8003)

should do in this context compared to celastrol.

7. At lines 274-276, the authors suggest that mutation of Lys536 in Nur77 impairs the effect of celastrol on inducing their interaction with p62. Lys536 is involved in ubiquitination and ubiquitination is hypothesized to be involved in recruiting p62. However, it is not clear at this point that celastrol mediates this interaction or that is essential for the phase separation of these components. Indeed, mutations of the Lys does not impact the phase separation of p62/Nur77. Only in the subsequent lines the authors detail that Nur77-LBD has specific interaction with the p62 in presence of celastrol and that mutation of the Lys suppress the phase separation. This perhaps even suggests that the celestrol interactions is mediated by ubiquitin or that body celestrol and ubiquitin interactions are required for phase-separation to occur. These observations are very important to define the multivalence of the states that are studied.

8. At lines 180-186 as well as in the paper figures, FRAP data are reported in terms of half time and or percentage of recovery after a given lag time. This is sort of confusing, since it does not allow a one-to-one comparison. A “ $T(1/2)$ ” can be obtained also for measurements that do not recover completely and will represent the half-time of recovery of the dynamic part. The missing amplitude will represent the fraction of molecules not diffusing or diffusing on a much slower time scale. In this respect, instead of reporting a % of recovery after different lag times, it seems like it would be more significative to choose an identical lag time for all the measurements and compare the percentage of amplitude at that time.

9. At line 210 the authors states that they tested for whether the droplets were irreversible aggregates or reversible phase-separated condensates. The dilution and observation of smaller objects does not disprove the formation of irreversible or long-standing aggregates, since the smaller objects could well be the remaining aggregate. The proper test would consist in titrating decreasing concentrations of protein against a fixed crowder (e.g. PEG of a given size) concentration and identify the threshold for phase separation. Then take a solution that contains

performed droplet and dilute down the protein concentration (but not the polymer concentration) below the threshold concentration and show that there is no phase-separation. This partially performed at in extended figure 3d and, based on those observations, it would suffice to add 1 or 2 further dilutions in Ext.Fig. 3e to pass the threshold (about 0.1 μM in PEG 3.35k in panel d, possibly not the same in PEG 8k panel e).

10. In extended figures 1 and 2 the fluorescence intensities should have an adequate x-axis. Indication that the scale bar is 10 μm (there is a typo in the legend saying 10 μM) would provide an important information for comparison with other objects of analogous nature as well to enable comparison with simulations.

11. The sentence at line 218-220 reads odd: "revealed that it exhibited as a single spherical particle". It is not clear the subject of the sentence and the objects do not appear to be perfectly spherical.

12. Similarly, the sentence at 227-228, "we found that... displays the highest disorder tendency in a broader IRD lacking well-defined structure" can perhaps be simplified to be "we found that... exhibits the highest structural disorder propensity".

Response to Reviewer 1

We appreciate the reviewer's time and effort in evaluating our manuscript and valuable comments provided to improve our paper. The major concerns of Reviewer 1 relate to the physiological significance of Nur77- and p62-mediated mitophagy and additional evidence supporting the Nur77/p62-mediated mitophagy. To address the physiological role of Nur77-mediated mitophagy, we provide a set of data illustrating the importance of Nur77-mediated mitophagy in the liver during aging (**Figure 1**, *Fig. 1d–h and Supplementary Fig. 1h–o in the revised manuscript*). These data demonstrated that the expression of Nur77, its colocalization with mitochondria and p62, and the mitophagy of the liver decline during aging and that Nur77^{-/-} mice displayed reduced mitophagy in the liver of both young and aged mice. To confirm the mitophagic effect of Nur77 and p62 and their interaction, we provide new data showing decreased mitophagic effect of celastrol in autophagy-deficient cells (**Figure 4**, *Supplementary Fig. 1c in the revised manuscript*), a set of new electron microscopy (EM) data demonstrating the localization of Nur77/p62-anchored mitochondria in mitophagosome and autolysosome (**Figure 5**, *Fig. 1c in the revised manuscript*), and immunostaining results illustrating the colocalization of Nur77/p62 condensates with early autophagy proteins WIP12 and ATG16L1 but not ULK1 and FIP200 in a phase separation-dependent manner (**Figure 2**, *Supplementary Fig. 6b–6e in the revised manuscript*). As requested, we have conducted experiments to reveal the difference between Nur77 and FIP200 on binding p62 (**Figure 3**). Our manuscript is also revised to address other minor concerns raised by Reviewer 1 and concerns raised by Reviewers 2 and 3 (see our responses to Reviewer 2 and 3).

Our detailed responses to the Reviewer's concerns follow (Reviewer's comments are italicized; figures indicating data presented in this response are bolded; figures referring those shown in original and revised manuscripts are italicized and indicated).

Reviewer #1 (Remarks to the Author):

In this manuscript, Peng et al., demonstrate that phase-separated Nur77 increases fluidity and size of p62 in a its ubiquitination dependent manner to induce celastrol-dependent mitophagy. The study is well-conducted, and the manuscript is written clearly. Liquid-liquid phase separation (LLPS) in selective autophagy is attracting interest in autophagy-research field, and this study provides a novel insight of LLPS into regulation of the mitophagy. Major concerns are lacking a presentation of physiological significance of Nur77- and p62-mediated mitophagy and analyses of autophagy-related proteins on Nur77- and p62-condensates.

Major comments

1. Throughout the manuscript, the authors utilized a triterpenoid antioxidant compound, celastrol to induce the mitophagy. What physiological conditions induces the Nur77- and p62-mediated mitophagy? Namely, what is a physiological role of Nur77- and p62-mediated mitophagy?

Response: Reviewer is concerned about the physiological significance of Nur77/p62-mediated mitophagy that we described in the manuscript. To address the concern, we studied the role of Nur77 and p62 in the mitophagy of the liver during aging. The aging process in the liver is driven by dysregulation of mitochondrial function and nutrient sensing pathways, which leads to cellular senescence and low-grade inflammation¹. Increasing evidence reveals a critical role of mitophagy in counterbalancing age-related pathological conditions². We showed previously³ that the mitophagic effects Nur77 is required to alleviate hepatic low grade inflammation in obesity that is known to accelerate aging of the liver⁴. These observations prompted us to hypothesize that Nur77/p62 play a role in the hepatic mitophagy during aging. Therefore, the liver of young (8 weeks old) and aged (2 years old) of mice and the corresponding Nur77^{-/-} mice were analyzed to determine the physiological role of Nur77-mediated mitophagy.

Our confocal immunofluorescence microscopy of liver section from young and aged mice revealed that the dynamic nature of hepatic mitochondria was altered during aging, in which the size of mitochondria in the liver of aged mice was significantly increased (**Figure 1a–1c**, *Fig. 1d,e in the revised manuscript*), in agreement with previous reports that enlarged mitochondria termed megamitochondria occur in aged liver⁵. The increase in the size of mitochondria in aged mice was accompanied with liver enlargement (**Figure 1d, 1e**, *Supplementary Fig. 1h, i in the revised manuscript*) and increased hepatic inflammation (**Figure 1f**, *Supplementary Fig. 1j in the revised manuscript*). A recent study showed that alteration of p62-mediated Parkin-independent mitophagy contributes to the formation of megamitochondria in the liver⁶. The role of Nur77 in mediating the mitophagy of the liver during aging is illustrated by our data showing that knocking out Nur77 significantly increased the size of hepatic mitochondria both in young and aged mice (**Figure 1b, 1c**, *Fig. 1d,e in the revised manuscript*). In addition, the expression of Nur77 decreased during aging (**Figure 1g**, *Fig. 1f in the revised manuscript*). These results reveal a critical role of Nur77 in regulating mitochondrial architecture during aging. In studying Nur77-mediated mitophagy, we found that Nur77 colocalization with mitochondria (indicated by Hsp60 staining) and p62 was significantly reduced during aging (**Figure 1h–1j**, *Supplementary Fig. 1k–m in revised manuscript*). The colocalization of LC3 with mitochondria in the liver was also decreased (**Figure 1k, 1l**, *Supplementary Fig. 1n, o in the revised manuscript*), suggesting a decline of Nur77/p62-mediated mitophagy during aging. Knocking out Nur77 caused defective mitophagy in the liver as Nur77^{-/-} mice displayed reduced colocalization of LC3 with mitochondria in the liver sections from both young and aged mice. To provide more direct evidence supporting the role of Nur77 in mitophagy during aging, we delivered AAV-9 plasmid that expresses the mitophagy biosensor EGFP-mCherry-COX8 to the livers of young and old mice by intravenous injection. Two weeks after injection, mCherry-COX8 exposure was detected (indicated by red color) in the liver of young control mice but not the young Nur77^{-/-} mice (**Figure 1m, 1o**, *Fig. 1g, h in the revised manuscript*). The mCherry-COX8 exposure was not detected in the liver of aged control and Nur77^{-/-} mice, likely due to decreased Nur77 expression in aged mice. Together, these results demonstrate that Nur77-mediated mitophagy plays a role in aging, addressing the physiological role of Nur77/p62-mediated mitophagy.

Figure 1. Nur77 is necessary for liver mitochondrial autophagy.

a. Representative images show Hsp60, a mitochondrial marker, in the liver tissue from wild-type and Nur77^{-/-} mice in aging model. Young mice, 8 weeks old. Aged mice, 2 years old.

b, c. The statistical analysis of mitochondrial size was represented as mean \pm SEM. **P<0.01, ***P<0.001, ****P<0.0001. N.S. no significance. Scale bar, 10 μ m.

- d.** The body weight from wild-type and Nur77^{-/-} mice in aging model. Statistical analysis of body weight was represented as mean ± SEM of five mice. N.S. no significance.
- e.** The ratio of liver weight and body weight from wild-type and Nur77^{-/-} mice in aging model. Statistical analysis of the ratio was represented as mean ± SEM of five mice. *P<0.05, **P<0.01. N.S. no significance.
- f.** Immunostaining detection of inflammatory cytokine CD68 in wild-type and Nur77^{-/-} mice in aging model. Scale bar, 25 μm.
- g.** The expression of Nur77 protein in the liver tissue from wild-type and Nur77^{-/-} mice in aging model.
- h-j.** Representative images show colocalization of Nur77 with p62 and Hsp60 in young and aged mice. Pearson's correlation coefficients of Nur77 with p62 and Hsp60 were also shown. Statistical analysis was performed with unpaired/two-tailed T-tests. ***P<0.001. Scale bar, 10 μm.
- k-l.** Representative images illustrate colocalization of LC3 with Hsp60 in the livers from wild-type or Nur77^{-/-} mice in aging model. Statistical analysis of Pearson's correlation coefficients of LC3 with Hsp60 were shown. **P<0.01. N.S. no significance. Scale bar, 10 μm.
- m.** Representative images of EGFP-mCherry-COX8 in the liver from wild-type or Nur77^{-/-} mice in aging model. Scale bar, 2 μm.
- o.** Cells that show mCherry-COX8 accumulation on liver tissue are quantified. Values represent the mean ± SEM (n = 5–6 mice). ****P<0.0001. N.S. no significance. Scale bar, 2 μm.

2. In the case of selective autophagy for Ape1 complex, P-body, and stress granule, liquidity or interacting protein(s) have been shown to be crucial factors for the structures to be surrounded by an autophagosome (Yamasaki A, et al. Liquidity Is a Critical Determinant for Selective Autophagy of Protein Condensates. Mol Cell 77, 1163-1175 e1169 (2020), Zhang G, Wang Z, Du Z, Zhang H. mTOR Regulates Phase Separation of PGL Granules to Modulate Their Autophagic Degradation. Cell 174, 1492-1506 e1422 (2018), Turakhiya A, et al. ZFAND1 Recruits p97 and the 26S Proteasome to Promote the Clearance of Arsenite-Induced Stress Granules. Mol Cell 70, 906-919 e907 (2018)). To prove whether the fluidity and size of Nur77- and p62-positive condensates determine autophagosome formation around mitochondria, the authors should investigate the localization of core ATG proteins essential for autophagosome formation such as FIP200, ULK1, WIPI2, ATG16L1 and LC3 on the condensates. And, also should verified failure of such localization in the case of Nur77-mutants including K567R and ΔIDR.

Response: Reviewer asks more analysis on the colocalization of Nur77- and p62-positive condensates with core ATG proteins. We showed in our original manuscript that the Nur77/p62 condensates could colocalize (*Supplementary Fig. 2 in the original manuscript*) and interact with LC3 (*Fig. 6c, d in the original manuscript*). As suggested, we conducted immunostaining assays to study whether the Nur77/p62 condensates colocalized with other core ATG proteins including FIP200, ULK1, WIPI2, ATG16L1. Our results showed that ULK1 and FIP200 failed to colocalize with celastrol-induced p62 condensates formed with Nur77 or Nur77-mutants (Nur77-K536R and Nur77-ΔIDR) (**Figure 2a and 2b, Supplementary Fig. 6b, c in the revised manuscript**). In contrast, WIPI2 and ATG16L1, similar to LC3, exhibited extensive colocalization with the p62 condensates formed with Nur77 but not Nur77 mutants (Nur77-K536R and Nur77-ΔIDR) (**Figure 2c and 2d, Supplementary Fig. 6d, e in the revised**

manuscript). These results demonstrated that Nur77/p62-mediated mitophagy is independent on the ULK1/FIP200 complex but requires WIPI2, ATG16L1, and LC3. They also revealed the critical role of Nur77 ubiquitination and its N-terminal IDR in mitophagy.

Figure 2. Analysis of the colocalization of ATG proteins with p62 condensates formed with Nur77 or mutants in celastrol-induced mitophagy.

a–d. Representative images show colocalization of the indicated ATG protein with p62, Nur77, or mutants in the presence or absence of celastrol. Pearson's correlation coefficients of Nur77 or mutants with p62 and ULK1 were also shown. Statistical analysis was performed with unpaired/two-tailed T-tests. *** $P < 0.001$. N.S. no significance. Scale bar, 10 μm .

3. The formation of the autophagosome around phase-separated p62 requires sequential and antagonistic steps. First, a complex between p62 and FIP200 is formed to initiate the selective autophagy of p62. This complex is mutually exclusive to the one formed between p62 and LC3, indicating that the binding of p62 to LC3 occurs after dissociation from FIP200 (Turco E, et

al. FIP200 Claw Domain Binding to p62 Promotes Autophagosome Formation at Ubiquitin Condensates. Mol Cell 74, 330-346 e311 (2019).) The authors should investigate whether Nur77 have an effect on the interaction.

Response: Reviewer asks to examine whether Nur77 has an effect on the sequential and antagonistic steps of phase-separated p62, in which p62 interaction with FIP200 is replaced by LC3 for the formation of the autophagosome around phase-separated p62 body. Such a sequential and antagonistic steps of phase-separated p62 is required for the autophagy of protein aggregates. Whether it is also required for p62-mediated formation of autophagosome around targeted mitochondria remains unknown. The basis of the sequential and antagonistic steps of p62-mediated autophagy is the competitive binding of FIP200 and LC3 to the same LIR motif of p62. However, in Nur77-mediated autophagy of mitochondria, Nur77 binding to p62 is mediated by the UBA domain and PB1 domain but not the LIR motif of p62. Thus, we consider it unlikely that Nur77-dependent formation of the autophagosome by phase separated p62 involves the sequential and antagonistic steps of FIP200 and LC3. This is supported by our previous finding that Nur77 could form a complex with p62 and LC3 (*Fig. 6c, d and Supplementary Fig. 2 in the original manuscript*). To further support our notion, we conducted experiments to study the effect of Nur77 on FIP200 interaction with p62. Our co-immunoprecipitation assays demonstrated that transfecting Nur77 had no apparent inhibitory effect on the interaction between p62 and FIP200 in the absence or presence of celastrol or wortmannin that blocks autophagy before isolation membrane elongation (**Figure 3a**). Knocking down Nur77 also failed to affect the interaction between p62 and FIP200 (**Figure 3b**). These results are consistent with the lack of FIP200 colocalization with p62 or Nur77 during celastrol-induced mitophagy shown above (**Figure 2b, Supplementary Fig. 6c in the revised manuscript**). Thus, Nur77-dependent formation of the autophagosome for mitophagy by phase separated p62 does not involve the FIP200-mediated sequential and antagonistic steps.

Figure 3. Nur77 does not affect p62 interaction with FIP200.

a. HeLa cells transfected with the indicated expression vectors were treated with or without wortmannin (1 μ M) and celastrol (2 μ M) for 1 h, and analyzed by co-IP assay for the interaction between FIP200 and p62.

b. HeLa cells transfected with Nur77 siRNA and the indicated expression vectors were treated with or without wortmannin (1 μ M) for 1 h, and analyzed by co-IP assay for the interaction between FIP200 and p62.

4. To show autophagy-dependency, the authors should show the impairment of the celastrol-dependent mitochondria degradation in autophagy-deficient cells.

Response: Reviewer requests data to illustrate autophagy dependent degradation of celastrol-induced mitophagy by using autophagy-deficient cells. This is a good suggestion, which will further support the mitophagic effect of celastrol. Accordingly, we used si-RNA to knock down the autophagy protein Atg7 in HeLa cells. EGFP-mCherry-COX8 reporter assay showed that the induction of mitophagy by celastrol was decreased in cells transfected with Atg7 siRNA, which inhibited Atg7 expression. Our fluorescence-activated cell sorting (FACS)-based dual color fluorescence-quenching assays revealed a significant reduction of celastrol-induced mitophagy (from 9.81% to 0.52%) when the expression of Atg7 was knocked down by Atg7 siRNA transfection (**Figure 4, Supplementary Fig. 1c in the revised manuscript**). Thus, celastrol-induced degradation of mitochondria is autophagy-dependent.

Figure 4. Celastrol-induced mitophagy is dependent on Atg7.

a. HeLa cells transfected with control (Si-NC) or Atg7 siRNA (Si-Atg7) and analyzed for celastrol induction of mitophagy by EGFP-mCherry-COX8 assay as described in Methods. Lower panel shows the expression of Atg7 protein in HeLa cells transfected with control and Atg7 siRNA. Scale bar, 10 μm.

b. Quantification of celastrol-induced mitophagy in HeLa cells transfected with control or Atg7 siRNA by flow cytometry. Statistical analysis was performed with unpaired/two-tailed T-tests. Data are shown as mean ± SEM. ***P<0.0001, N.S. no significance.

5. p62-condensates could be identified by their unique morphology by electron microscopy (EM) (Jakobi AJ, et al. Structural basis of p62/SQSTM1 helical filaments and their role in cellular cargo uptake. Nat Commun 11, 440 (2020).). The authors can show fine structure of Nur77- and p62-condensates around mitochondria by EM and compare them with p62-structures formed under stress condition. They may be also able to observe how isolation membranes sequester the mitochondrion.

Response: Reviewer asks to provide fine structure of Nur77- and p62-condensates around mitochondria by electron microscopy (EM). Accordingly, we performed immunogold EM to examine the localization of the Nur77 and p62 condensates around mitochondria. Dual immunogold staining revealed no detectable Nur77 or p62 on healthy mitochondria in control cells. When cells were treated with celastrol, however, damaged mitochondria were densely decorated at their outer membrane with anti-p62 (10 nm) and anti-Nur77 (15 nm) immunogold labels, which could be found within the autophagosome or autolysosome (**Figure 5a**, *Fig. 1c in the revised manuscript*). Dual immunogold staining also showed that LC3 (15 nm) and p62 (10 nm) immunogold particles were colocalized near mitochondrial out membrane, which again were engulfed by autolysosome (**Figure 5b**, *Fig. 1c in the revised manuscript*). These EM results further confirm the role of Nur77/p62 condensates in mediating celastrol-induced mitophagy. Due to time limitation and technical challenging, we have not been able to observe how isolation membranes sequester the mitochondrion, which will be followed in the future.

Figure 5. Nur77 and p62 colocalize with the mitochondrial outer membrane during celastrol-induced Mitophagy.

a. Electron micrographs of HeLa cells stained with 15 nm immunogold-conjugated Nur77 antibody to detect Nur77 (red), and 10 nm immunogold-conjugated p62 antibody to detect p62 (green). Cells were treated for 1 hr with celastrol. The blue dotted line indicates autophagosome. Mito, mitochondrion, Scale bar, 200 nm.

b. Electron micrographs of HeLa cells stained with 15nm immunogold-conjugated LC3 antibody to detect LC3, and 10 nm immunogold-conjugated p62 antibody to detect p62. Cells were treated for 1 hr with celastrol. The blue dotted line indicates autophagosome. Mito, mitochondrion, Scale bar, 200 nm.

6. In Figure 5b, the quantity of Nur77 to bind to p62 dramatically decreased by K536R mutation. Nevertheless, the K536R mutation did not affect co-localization efficacy with p62 (Fig. 5d).

Response: Reviewer is concerned that although binding of Nur77-K536R to p62 was reduced compared to wild-type Nur77 revealed by coimmunoprecipitation assay (*Fig. 5b in the original and revised manuscript*), the K536R mutation did not affect Nur77 colocalization with p62 (*Fig. 5d in the original manuscript*). We would like to clarify that the wild-type Nur77 showed almost 100% colocalization with p62 in the cytoplasm when cells were treated with celastrol. However, the Nur77-K536R mutant only displayed partial colocalization with p62, with a significant amount of the mutant protein failed to colocalize with p62 (indicated by white arrows) (**Figure 6**). This is in agreement with our co-immunoprecipitation result (*Fig. 5b in the original manuscript*) and reveals the importance of K536 of Nur77 in its binding to p62. Although the binding and colocalization between Nur77-K536R and p62 are reduced, the size of condensates formed by Nur77-K536R and p62 is almost the same as that formed by Nur77 and p62, further strengthening our conclusion that the IDR of Nur77 is responsible not only for the binding of Nur77 to p62 but also for the expansion of the Nur77/p62 condensates.

Figure 6. The role of Nur77 ubiquitination in its interaction with p62.

Immunofluorescence images showing the effect of Nur77 ubiquitination on mCherry-p62 condensate formation after treatment with or without celastrol. White arrows indicate GFP-Nur77-K536R particles. Scale bar, 10 μm .

7. In Figure 7a, distinct to Fig. 5b, the interaction of p62 with Nur77 in normal conditions was not detected. Nevertheless, the authors claimed that “Removing UBA domain impair celastrol-dependent but not independent interaction”. The bands corresponding to Nur77 were broad and obscure.

Response: Reviewer questions our statement that “Removing UBA domain impair celastrol-dependent but not independent interaction” as judged by lack of the interaction between p62 and Nur77 in normal conditions shown in Fig. 7a. We have consistently observed celastrol-independent interaction between Nur77 and p62, involving the IDR of Nur77 and the PB1 of p62. We apologize that a shorter exposure blot was shown in Fig. 7a, which has now been replaced with a longer exposed one (**Figure 7, Fig. 7b in the revised manuscript**). The new immunoblot is comparable to that shown in Fig. 5b, illustrating the celastrol-independent interaction between Nur77 and p62 in normal conditions. Reviewer is also concerned about broad and obscure Nur77 bands shown in Fig. 7a. In this figure, we ran gel electrophoresis for a longer time in order to separate different p62 mutants analyzed in this figure, which made Nur77 bands broad and fuzzy. However, it does not affect our conclusion.

Figure 7. Analysis of Nur77 interaction with p62 and mutants.

HeLa cells transfected with Myc-Nur77 and the indicated Flag-p62 or mutants were treated with or without celastrol, and analyzed for Nur77 interaction with p62 and mutants by co-IP assay.

8. The immunoblot in Figure 7b should be also replaced with representative results.

Response: Reviewer asks to replace the immunoblot in Fig. 7b. We have now replaced the figure with a new immunoblot that clearly illustrates celastrol-independent interaction between Nur77-IDR and p62 (**Figure 8, Fig. 7c in the revised manuscript**).

Figure 8. Celastrol independent interaction between Nur77-IDR and p62.

HeLa cells transfected with the indicated expression vectors were treated with or without celastrol, and analyzed for Nur77-IDR interaction with p62 and p62- Δ PB1 by co-IP assay.

9. *Supplementary Figure 1a: in the text (Figure legend), the authors described “TNF- α -treatment”. But, in the Figure, the data is missing. Mislabeled?*

Response: We apologize for the mislabeling of *Supplementary Fig. 1a* (*Supplementary Fig. 1e in the revised manuscript*), which has now been corrected.

10. *Supplementary Figure 7a: immunoblot data should be replaced with representative one.*

Response: We believe that this critique relates to *Supplementary Fig. 7b*, which has now been replaced with a new representative one (**Figure 9**, *Supplementary Fig. 7b in the revised manuscript*).

Figure 9. DBD domain of Nur77 is dispensible for Nur77 interaction with p62.

HeLa cells transfected with the indicated expression vectors were treated with or without celastrol, and analyzed for p62 interaction with Nur77 and Nur77- Δ DBD by co-IP assay.

11. *Recent study showed that overexpression of NBR1 blocks the autophagic turnover of p62-structures (Sanchez Martin P, Sou YS, Kage yama S, Koike M, Waguri S, Komatsu M. NBR1-mediated p62-liquid droplets enhance the Keap1-Nrf2 system. EMBO Rep 21, e48902 (2020)). The quantity of NBR1 bound to p62-structures might regulate fluidity and size of p62 and in turn affect the direction of autophagic engulfment on the p62-condensates. The binding Nur77 to p62 might be competitive to NBR1-binding. The authors may discuss this point.*

Response: p62 and NBR1 share a common domain organization. They interact with each other through their PB1 domains and act cooperatively in different forms of selective autophagy⁷. A recent study showed that overexpression of NBR1 blocks the autophagic turnover of p62-structures by promoting the fluidity and size of p62⁷. As both NBR1 and Nur77 bind to the

PB1 domain of p62 and increase the fluidity and size of p62, Reviewer asks to discuss whether Nur77 and NBR1 may bind to p62 competitively. This is an excellent suggestion given the fact that NBR1 blocks autophagic engulfment of the p62-condensates while Nur77 serves to promote their connection to the autophagic machinery. As suggested, we have now discussed the possible effect of NBR1 in Nur77-mediated mitophagy in the revised manuscript.

12. *The pioneer studies showing the roles of p62 in selective autophagy should be cited. Bjorkoy G, et al. p62/SQSTM1 forms protein aggregates degraded by autophagy and has a protective effect on huntingtin-induced cell death. J Cell Biol 171, 603-614 (2005). Komatsu M, et al. Homeostatic levels of p62 control cytoplasmic inclusion body formation in autophagy-deficient mice. Cell 131, 1149-1163 (2007).*

Response: We apologize for missing these two important pioneer studies, which have now been cited in the revised manuscript.

Response to Reviewer 2

We thank Reviewer's time and effort in evaluating our manuscript, and appreciate very much reviewer's valuable comments and suggestions to improve our paper. The main concerns of the Reviewer relate to insufficient description of celastrol, the presentation of data and their statistical analysis, and the generality of our described mechanism. To address the concerns, we have provided more description of celastrol, improved our data presentation with statistical analysis, and included new figures revealing a critical role of the IDR of two other members of the Nur77 (NR4A1) family, Nurr1 and NOR1, in promoting their body formation in cells (**Figure 16**, *Supplementary Fig. 4c-f in the revised manuscript*) and the ability of BI1071, another Nur77 binder, in inducing the phase separation of Nur77 (**Figure 13**, *Fig. 4g in the revised manuscript*). Our manuscript is also revised to address other minor concerns and concerns raised by Reviewers 1 and 3 (see our responses to Reviewers 1 and 3).

Below are our detailed responses (Reviewer's comments are italicized; figures indicating data presented in this response are bolded; figures referring those shown in original and revised manuscripts are italicized and indicated).

Reviewer #2 (Remarks to the Author):

Peng et al., have presented a very interesting study following their previous work where they have discovered that mitochondrial Nur77 can mediate autophagy of dysfunctional mitochondria upon binding to celastrol. Here they demonstrate novel mechanistic insights on how Nur77 and p62 can coordinate (via phase separations) to sequester damaged mitochondria and to connect targeted cargo to the autophagic machinery. It is a very interesting and well written manuscript. I do however have a couple of concerns that need to be addressed.

Major critiques

1) The authors should provide some more information on Celastrol and its use. They only mention that it is "a potent anti-inflammatory triterpenoid quinone-methide". They should provide more information, is it often clinically used? To treat which cases? If known: what are known effects and on which cells?

Response: We apologize for the insufficient information provided in the manuscript regarding the nature and usage of celastrol. Celastrol is isolated from *Trypterygium wilfordii* Hook F. (commonly called the thunder god vine), a Chinese medicinal herb that has been used widely and successfully for centuries for treating number of autoimmune and inflammatory diseases including rheumatoid arthritis and lupus due to its potent anti-inflammatory effect⁸. Celastrol has recently drawn considerable interest, after the discovery that the compound possesses potent anti-obesity activity⁹. However, the transformation of this traditional medicine into modern drug requires our identification of intracellular targets and understanding of its mechanism of action. We recently reported that the anti-inflammatory effect of celastrol is due to its binding to Nur77, which triggers Nur77-dependent elimination of dysfunctional mitochondria by autophagy³. This finding laid the groundwork for the current study, which

aims to investigate the mechanism by which Nur77 interaction with p62 mediates celastrol-induced mitophagy. We have now provided a more detailed description of celastrol in the revised manuscript.

2) *Figure 1A*: the authors also should provide a zoom inset of the control cells. Are the microscopy settings the same in control cells and stimulated cells? The fluorescence intensity of the control seems to be less. *Figure 1B and D*: Next to the FACS plots, the authors should provide a quantification of at least 3 independent experiments and perform proper statistics to show variability and reproducibility of the experiment. In addition, they should provide example plots of the gating strategy (were the doublets and dead cells excluded?). The materials and methods section also should contain a paragraph describing the flow cytometry experiments.

Response: As requested, we have now provided a zoom inset of all control cells shown in **Figure 10a and 10c** (*Fig. 1a, b in the revised manuscript*). In addition, we have corrected the fluorescence intensity for easy comparison. For the statistical significance of Figures 1b and d, quantitation of 3 independent experiments and their statistics are provided (**Figure 10b and 10d**, *Supplementary Fig. 1d, e in the revised manuscript*). Example of plots of the gating strategy is included as *Supplementary Fig. 1a* (**Figure 10e**, *Supplementary Fig. 1a in the revised manuscript*). Flow cytometry experiments are now described in the Materials and Methods section.

Figure 10. Nur77 and p62 are required for celastrol-induced mitophagy.

a, b. Representative images and quantification of celastrol-induced mitophagy in HeLa, Nur77^{-/-} HeLa, and Nur77^{-/-} HeLa cells transfected with Myc-Nur77 by EGFP-mCherry-COX8 assay as described in

Methods. Statistical analysis was performed with unpaired/two-tailed T-tests. N=3. Data are shown as mean \pm SEM. ****P<0.0001, N.S. no significance. Scale bar, 10 μ m.

c. Representative images of celastrol-induced mitophagy in MEFs and p62^{-/-}MEFs by EGFP-mCherry-COX8 assay as described in Methods. Scale bar, 10 μ m.

d. Quantification of celastrol-induced mitophagy in HeLa cells transfected with control or p62 siRNA by flow cytometry. Above panel shows the efficacy in knocking down p62 by p62 siRNA transfection. Statistical analysis was performed with unpaired/two-tailed T-tests. N=3. Data are shown as mean \pm SEM. ****P<0.0001, N.S. no significance.

e. Gate setting strategy of FACS plots of EGFP-mCherry-COX8 assay.

3) *Figure 2A and B: The pictures of these figures are stills of extended video 1. Data from multiple cells should be shown. Quantification of the number and the size of the vesicles per cell would provide additional insight and will show the increase over time (including variability and reproducibility of the experiment). From these stills it is difficult to see what the increase over time is, especially since already more than 5 puncta are present at 0 min celastrol.*

Important: The quantification should not be done on overexposed pictures, as shown for the GFP channel in these figures. (same is true for extended figure 1b, there is overexposure of GFP-nur77 and mCherry-p62, therefore it is not possible to correctly plot their fluorescence intensities)

Line 167: "the condensates of p62 and LC3 had a greater diameter compared with those of Nur77 and mitochondria" to make this statement, quantification of the diameter of multiple droplets per cell and multiple cells should be done including proper statistics. Additionally, a color swab needs to be performed to exclude diffraction effects. Additionally, quantification of diameter of vesicles should not be done on overexposed samples as shown in Supplementary fig2a and 2b

Figure 2C, why is the GFP channel in the control much less intense compared to the celastrol stimulated cells? To be able to properly do the quantification as the authors have shown in figure 2c in the Nur77^{-/-}-HeLA+GFP-Nur77 experiment, similarly exposed pictures should be compared.

Response: We appreciate the Reviewer's valuable suggestions and comments to improve our data presentation and their interpretation. Accordingly, we have conducted the requested experiments and modified our figures and presentation as described below.

Regarding the concern about the real-time images showing the formation and fusion of GFP-Nur77 and mCherry-p62 droplets after treatment with celastrol for 1 hr, we apologize for the confusing labeling of the stills (*Fig. 2a and 2b in the original manuscript*). 0 min in the figure indicates the beginning of video recording after cells were treated with celastrol for 1 hr. That is why there are already 5 puncta present at 0 min of the recording. GFP-Nur77 is exclusively localized in the nucleus before celastrol treatment. These stills of video together with videos showed that Nur77/p62 condensates were formed through fusion of either undetectable micro-sized droplets or detectable droplets after cells were treated with celastrol for 1 hr. We have now modified our labeling to make it easier for readers to understand (*Fig. 2b in the revised manuscript*). In addition, we have provided a new figure with quantification of the number and

the size of droplets (**Figure 11a**; *Fig. 2a in the revised manuscript*) to illustrate the exclusive nuclear localization of Nur77 before celastrol treatment and the effect of celastrol on inducing the formation and fusion of cytoplasmic GFP-Nur77 and mCherry-p62 droplets. This new figure shows that the number of Nur77/p62 droplets decreases, while their size increases over time after celastrol treatment. With this new figure, Fig. 2b in the original manuscript is now placed as Supplementary figure (*Supplementary Fig. 2a in the revised manuscript*). With an addition of a new figure and revised labeling, we hope that the role of celastrol on inducing the formation and fusion of cytoplasmic GFP-Nur77 and mCherry-p62 droplets can be clarified.

Regarding the concern that data from multiple cells should be shown, we could not show many cells in one field because the high magnification of images is needed. We have shown in original Fig. 2a, b and many other figures throughout the manuscript that celastrol treatment results in the formation of cytoplasmic condensates, while Nur77 is predominantly nuclear in the absence of treatment. This has been consistently seen with majority of cells studied. To address the concern, we have provided another real-time images showing the effect of celastrol on inducing the fusion and expansion of Nur77/p62 droplets (**Figure 11b**; *Supplementary Fig. 2b and Supplementary video 3 in the revised manuscript*).

Reviewer also requests quantification of the diameter of multiple droplets per cell and multiple cells with proper statistics to support our conclusion that the diameter of p62 and LC3 condensates is bigger than that of mitochondria. We have replaced our overexposed pictures (*Supplementary Fig. 2a, b in the original manuscript*) with new pictures with reduced exposure (**Figure 11c and 11d**; *Supplementary Fig. 2c, d in the revised manuscript*). The new figure also shows the quantification of the diameters of multiple droplets per cells and multiple cells including statistical analysis. To exclude diffraction effects, color swab was conducted. With these suggested modifications, our results are now more convincing, which clearly demonstrate the engulfment of mitochondria by p62- and LC3-containing autophagosomes.

Regarding the concern about different GFP channel intensities of control and celastrol stimulated cells shown in Fig. 2c, similarly exposed pictures are now used (**Figure 11e**, *Fig. 2c in the revised manuscript*). Comparison and quantitation of the size and number of Nur77/p62 condensates again revealed a potent effect of celastrol on inducing the formation of the Nur77/p62 condensates.

Figure 11. Celastrol promotes Nur77/p62 condensate formation.

a. Representative images show time-dependent experiment under treatment with celastrol. Scale bar, 10 μm . Quantitative analysis of the number and size of Nur77/p62 condensate formation.

b. Real-time images showing formation and fusion of GFP-Nur77 and mCherry-p62 droplets in HeLa cells after treatment with celastrol (2 μM) for 1 hr as indicated by the arrows. Scale bar, 10 μm .

c, d. Celastrol-induced Nur77 condensates colocalize with p62, LC3 at mitochondria. Fluorescence intensity quantitative analysis of indicated condensates shows that mitochondria are engulfed by p62 and LC3 recruited by Nur77 to form condensates. Scale bar, 10 μm . * $P < 0.05$.

e. Representative images illustrating the role of celastrol in promoting the formation of p62 body in a Nur77-dependent manner. Scale bar, 10 μm .

4) Figure 3d, f, g, h reveals that Nur77 also forms puncta in the nucleus. Also Figure 2a, b and c (GFP-Nur77) should show this, but since these pictures are overexposed, this is not visible. Therefore in the whole manuscript only pictures should be shown where samples are not overexposed.

Response: Reviewer is concerned about our data illustrating Nur77 punctate formation in the nucleus (Figures 3d, f, g), and yet other data in the manuscript did not show similar nuclear Nur77 puncta. We have followed the suggestion to replace the overexposed pictures in the manuscript. We would like to clarify that Nur77 is a nuclear receptor of transcription factor, and it is predominantly residing in the nucleus in normal conditions. Our observed formation of Nur77/p62 condensates and their role in mediating celastrol-induced mitophagy involves only a small portion of Nur77. Thus, it is sometimes difficult for us to simultaneously show Nur77 puncta in both the nucleus and the cytoplasm.

5) Suppl figure 3c (30% PEG-3.35k) and 3d (5 μ M mCherry-Nur77); these picture show rather amorphous condensates than spherical droplets. Can the size of these condensates then properly be measured? Isn't it better to exclude these datapoints from the graphs?

Response: We appreciate Reviewer's suggestion on our Supplementary Fig. 3c. We have repeated these experiments using freshly prepared mCherry-Nur77 protein. Our new data show spherical Nur77 droplets in 30% PEG-3.35K and at 5 μ M concentration (**Figure 12, Supplementary Fig. 3c, d in the revised manuscript**).

Figure 12. Characterization of Nur77 condensates.

a. Effect of PEG concentration on Nur77 droplet formation. Concentration dependent effect of PEG-3.35K on mCherry-Nur77 droplet formation. Scale bar, 5 μ m.

b. Nur77 droplet formation is dependent on Nur77 concentration. Concentration dependent effect of Nur77 protein on droplet formation. Scale bar, 5 μ m.

6) The legend of figure 4 should include more details. Are the cells stimulated with celastrol in fig 4c? For 4c the authors also should quantify the diffuse, diffuse/punctate and punctate cells for these mutants in HeLa cells as they did for the purified mutants in extended figure 4. The authors also should show the variability between independent experiments. The DAPI channel is hardly visible and for 4c: the authors also state: “while deleting the DBD had little effect”:

why is this stated? This figure shows (n=1 cell) that there are no true spherical droplets. 4d: does nuclear export from the nucleus to the cytoplasm happen in all cells? Or in a proportion of the cells? Does this not happen in untreated cells? What is the difference? To be able to draw conclusions on this somehow this needs to be quantified.

4e: why was only an RXR ligand taken along, and not another Nur77 ligand, such as 6-MP, C-DIMs or cytosporone B? This could proof whether it is dependent on Celastrol or merely ligand binding to Nur77

Response: We apologize for the insufficient experimental details provided in the legend of Fig. 4. It has now been revised as suggested. Fig. 4c illustrates the cellular distribution patterns of Nur77 mutants in the absence of celastrol. As suggested, we have now included statistical analysis of the diffusion and aggregation state of Nur77 mutants and the variation between experiments (**Figure 13a**, *Fig. 4d, 4e in the revised manuscript*). In this figure, DAPI channel has been enhanced to clearly identify the nucleus. We agree that the statement “while deleting the DBD had little effect” is inaccurate as Nur77- Δ DBD displayed amorphous condensates rather than spherical droplets seen with wild-type Nur77. Thus, the statement has been revised accordingly.

Regarding the efficacy of celastrol induction of Nur77 nuclear export (Fig. 4d in the original manuscript), the effect is observed in a majority of cells. Our statistical results show that celastrol-induced nuclear export of transfected GFP-Nur77 occurs in 73% of cells (**Figure 13b**, *Fig. 4f in the revised manuscript*).

Regarding the concern whether other Nur77 binding agents affect Nur77 phase separation, we have included a set of new data showing that BI1071¹⁰, a compound derived from methylene-substituted diindolylmethane (C-DIM), could also promote the phase separation of Nur77 (**Figure 13c**, *Fig. 4g in the revised manuscript*). Together, our results demonstrate that the binding of Nur77 by its ligands could lead to its phase separation.

Figure 13. Ligands promote Nur77 phase separation.

a. Representative images and quantification of Nur77 mutants in HeLa cells. Scale bar, 10 μ m.

b. Quantification of GFP-Nur77 nuclear export after treatment with or without celastrol (2 μ M) in HeLa cells. Scale bar, 10 μ m. Statistical analysis was performed with unpaired/two-tailed T-tests. Data are shown as mean \pm SEM. ***P<0.0001, N.S. no significance.

c. Droplet formation of GFP-Nur77 and mutants in HeLa cells treated with the indicated compounds 1 hr (celastrol (2 μ M), BI1071 (0.5 μ M), K-80003 (2 μ M)). Left: Representative droplet images of transfected GFP-Nur77 and mutants. Enlarged view of inset is also shown. Scale bar, 10 μ m. Right panels, quantification of GFP-Nur77 and mutant diffusion or aggregation.

7) Schematic drawings for the mutants used in figure 4 (Nur77 and P62) would be helpful.

Response: We have provided schematic drawings for Nur77 and p62 mutants (Figure 14, Fig. 4b and 7a in the revised manuscript).

Figure 14. Structure domain of Nur77 and p62.

Schematic representation of Nur77 and p62. Intrinsic disorder tendency of Nur77. IDR, intrinsic disorder region; DBD, DNA-binding domain; LBD, ligand-binding domain. PB1, Phox/Bem1p protein-protein binding domain. ZZ, zinc-finger domain. TB, TRAF6 binding domain, UBA, ubiquitin-associated domain.

8) Extended figure 7; what is compared with the statistics in figure D?

Response: Comparisons for the statistics shown in Supplementary Fig. 7d are now indicated (Figure 15, Supplementary Fig. 7e in the revised manuscript).

Figure 15. Nur77 and its mutants regulate p62 phase separation.

Quantification showing the effect of Nur77 and mutants in regulating p62 body formation in Nur77^{-/-} HeLa cells. Statistical analysis was performed with unpaired/two-tailed T-tests. Data are shown as mean ± SEM. *P<0.05, **P<0.01, ***P<0.001, ****P<0.0001, N.S. no significance.

9) For clarity it would be better to include in all legends which statistical test was used, including the n. Not only in the materials and methods section.

Response: We have now provided how statistical data were obtained in each figure legend.

10) In the discussion line 475 the authors state: "IDR-mediated phase separation may represent an important mechanism utilized by nuclear receptor family members to regulate diverse biological activities." This indeed may be true. Did the authors compare the IDR of Nur77 (NR4A1) with its family members Nurr1 (NR4A2) and NOR1 (NR4A3)? And did they ever investigate whether nurr1 and NOR1 also respond to celastrol? I think the discussion misses a paragraph discussing this.

Response: This is an excellent suggestion. We therefore conducted experiments to address whether the IDR of other members of the Nur77 subfamily undergoes phase separation. We constructed mutants of nuclear receptor Nurr1 (NR4A2) and NOR1 (NR4A3). By using the prediction program PONDR (VSL2) (<http://www.pondr.com/>), we found that the N-terminal of Nurr1 and NOR1 also exhibit the highest structural disorder propensity, which is similar to Nur77 (**Figures 16a and 16b**, *Supplementary Fig. 4c,d in the revised manuscript*). We then tested whether Nurr1 and NOR1 underwent LLPS in vivo, and found that ectopically expressed GFP-Nurr1 or GFP-NOR1 formed punctate structures in the nucleus, which were almost identical to those formed by Nur77. Removal of IDR from Nurr1 (GFP-Nurr1- Δ IDR) or NOR1 (GFP-NOR1- Δ IDR) greatly reduced their ability to form nuclear droplets, while GFP-Nurr1-LBD and GFP-NOR1-LBD were diffused (**Figures 16c and 16d**, *Supplementary Fig. 4e, f in the revised manuscript*). These results demonstrated that the N-terminal IDR of Nurr1 and NOR1 is responsible for the formation of their phase-separated condensates. When the effect of celastrol was tested, we did not observe any difference in the subcellular distribution and phase separation of Nurr1 and NOR-1, demonstrating that celastrol does not affect the phase separation of Nurr1 and NOR1, likely due to its inability to bind to these receptor proteins. Thus, the effect of celastrol is Nur77 specific. With these new results, we have provided a new paragraph discussing the role of IDR in mediating the diverse biological effects of nuclear receptor family members

Figure 16. Nurr1 and NOR1 form condensates in vivo.

a, b, Intrinsically disordered tendency of Nurr1 and NOR1. IDR, intrinsically disordered region; DBD, DNA-binding domain; LBD, ligand-binding domain.

c, d, Droplet formation of GFP-Nurr1 and GFP-NOR1 or mutants in HeLa cells treated with or without celastrol (2 μM). Right: quantification of diffusion or aggregation of GFP-Nurr1, GFP-NOR1, and their mutants. Statistical analysis was performed with unpaired/two-tailed T-tests. Data are shown as mean ± SEM. N.S. no significance.

Response to Reviewer 3

We appreciate very much reviewer's time and effort in evaluating our manuscript and insightful comments to improve the quality and impact of our findings. The main concerns of Reviewer 3 relate to our data interpretation and insufficient information provided for nonspecialized readers. In the revised manuscript, we have modified our interpretations such as "two phase separations" "Nur77 nuclear export" and provided schemes of Nur77 and p62 and a graphic summary to improve the readability of our manuscript. Below are our detailed responses (Reviewer's comments are italicized; figures indicating data presented in this response are bolded; figures referring those shown in original and revised manuscripts are italicized and indicated).

Reviewer #3 (Remarks to the Author):

The work of Peng et al. investigates the phase separation of Nur77 mediated by celastrol and its involvement in mitophagy. The study demonstrates that Nur77 undergoes phase-separation when mixed with crowding agents in vitro, reproducing the experimental observation of liquid-like puncta formation inside the nucleus of cells. Most importantly, the study: i) dissects the interplay between Nur77 and p62 and their recruitment in phase separated compartments within the cell, ii) identifies the regions and modifications controlling their interaction (and consequently their phase separation). The observations are convincing, and the body of work is remarkable for scope and number of experimental conditions investigated. However, part of the data interpretation is problematic when framed in the context of phase separation and these aspects should be addressed to provide a consistent connection between the experimental observations and the underlying invoked physics. Finally, given the functional relevance and multi-layer structure of these membraneless compartments, I expect this publication can be of interest for a large audience. Therefore, I would suggest improving readability to help the non-specialized audience in the navigation of the large number of experimental conditions explored in the manuscript.

In details:

1. The manuscript would benefit of a scheme of the protein components and their architecture (as described in the introduction of the paper). This will ease the reader in remembering which domains of the proteins are involved in which interactions or modifications. This is done for Nur77 in Fig. 4 but not for p62. Similarly, the same scheme can be used across figures to identify which components are tested.

Response: This is a great suggestion. Schemes of Nur77 and p62 as well as their mutants used in the study are now included in the revised manuscript (**Figure 17**, Fig. 4b and 7a in the revised manuscript).

Figure 17. Schematic representation of Nur77 and p62 as well as their mutants.

IDR, intrinsically disordered region; DBD, DNA-binding domain; LBD, ligand-binding domain. PB1, Phox/Bem1p protein-protein binding domain. ZZ, zinc- finger domain. TB, TRAF6 binding domain. UBA, ubiquitin-associated domain.

2. Similarly, a graphic summary of the emerging model would help understanding the different steps controlling cellular localization of the components, phase-separation, modification, cargo binding, and how these steps can lead to mitophagy.

Response: We fully agree that a graphic summary in the paper will help readers understanding the complex phase separation of Nur77 and p62 and the process of celastrol-induced mitophagy. Thus, a model of Nur77/p62-mediated phase separation in mitophagy is provided in the revised manuscript (**Figure 18**; *Fig. 8 in the revised manuscript*), which illustrates how celastrol binding promotes the phase separation of Nur77 through ubiquitin modification, which recruits p62/SQSTM1 to sequester damaged mitochondria, and how it acts in concert with the interaction between the IDR of Nur77 and the PB1 domain of p62/SQSTM1 to connect cargo mitochondria to the autophagic machinery.

Figure 18. A model depicting how Nur77 phase separation participates in autophagy of dysfunctional mitochondria (mitophagy). The phase separation of Nur77 and p62/SQSTM1 triggered through their multivalent interaction sequesters damaged mitochondria and connect the target cargo to the autophagic machinery.

3. Statement like: “The Nur77/p62 condensates require an additional phase separation mediated by the interaction of N-terminal intrinsically disordered region” (line 117-120) and similar ones in the discussion section (line 434-436) are highly confusing. Unless Nur77

undergoes proteolysis and it is divided in two segments (IDR and LBD) within the cell, it is not simple to talk about “an additional phase separation”. The phase separation propensity (phase boundaries) and properties (e.g. fluidity) of fragments of the protein do not provide direct insights on the behavior of Nur77 phase separation propensity and properties, because for the full-length protein there will be not a distinct phase separation of the IDR or folded domain (giving rise to two phase separations), but the whole protein will participate into a single phase separation (even though multiple phases can coexist). There are particular cases in which a micro-phase separation can be invoked with specific domains of the protein partitioning in a specific phase (as for block co-polymers) but this is not supported by the presented data. Indeed, all the experiments point to the fact that both Nur77 and p62 are multivalent, with part of the multivalence aided by celastrol and ubiquitination. This is an important result that has to be stressed more.

Response: Reviewer is concerned about our statement of “an additional phase separation” involving the interaction of the IDR of Nur77 and the PB1 of p62. We fully agree that it is inappropriate to consider ubiquitination-mediated and IDR-mediated phase separations of Nur77 as two separate phase separations given the fact that Nur77 acts as a full-length protein. Thus, we have deleted such a statement in the revised manuscript.

4. At line 167, LC3 phase separation is mentioned in comparison to Nur77. Though previously introduced in the text, it was not obvious if the protein itself was undergoing phase separation together with p62. A sentence clarifying this point may ease the reader in following the experimental design. The corresponding outcome is also extremely important and parallels previous finding from other groups (e.g the Kriwacki lab on the organization of nucleophosmin inside of the nucleolus) on the formation of multi-shell condensates. A mention to previous literature in this respect would help interpreting the possible mechanisms leading to these types of condensates.

Response: Reviewer asks to introduce LC3 condensates in the text when we compared the LC3 condensates with Nur77 condensates. This is an excellent suggestion. Accordingly, we provide an introduction about the recruitment of LC3 by p62 condensates after p62 underwent phase separation through ubiquitin-induced oligomerization and cited related references. Such an introduction help readers understanding our experimental design and the significance of our results. The oligomerization of p62 through the PB1 domain generates higher avidity interactions with ubiquitin and LC3-coated surfaces^{11, 12, 13}. We have also followed the suggestion to discuss the Kriwacki work on the formation of multi-shell nucleolus condensates, whose formation by LLPS of nucleophosmin facilitates the initial steps of ribosome biogenesis¹⁴. Such modifications help readers understanding our experimental design and the significance of our results that demonstrates the role of Nur77/p62 phase separation in connecting cargo mitochondria to LC3-containing autophagosomal membrane for autophagy.

5. The authors observed that Nur77-IDR undergoes phase separation in cells, whereas Nur77-Delta-IDR, LBD, or DeltaDBD reduces or even suppress phase separation in absence of celastrol. Whereas it is clear that Nur77-IDR has an essential role in driving the phase-

separation, observations that Nur77-LDB colocalizes with Nur77-IDR droplets suggests that heterotypic interactions between Nur77 domains may play a role. In this respect, it would be helpful to cast these experiments in the light of testing the role of homotypic and heterotypic interactions across domains of Nur77.

Response: Reviewer suggests that there is a heterotypic interaction between Nur77 domains in light of our observation that Nur77-LBD colocalized with Nur77-IDR droplets (**Figure 19, Supplementary Fig. 4a, b in the revised manuscript**). The heterotypic interaction is quite common for nuclear receptor superfamily members¹⁵. We also reported that the heterotypic interaction involving N-terminal A/B domain and C-terminal LBD occurs for retinoid X receptor¹⁶. Such heterotypic interaction involving N-terminal IDR and C-terminal LBD was also reported for Nur77¹⁷, providing an explanation for the colocalization of Nur77-IDR and Nur77-LBD. The heterotypic interaction between Nur77 domains is now discussed in the revised manuscript.

Figure 19. Role of Nur77-IDR in mediating phase separation of Nur77.

a-b. Purified GFP-Nur77-IDR (2 μ M) was incubated with mCherry-Nur77-IDR, mCherry-Nur77-DBD, mCherry-Nur77-LBD and mCherry or GFP (2 μ M) as control for 10 min in 10% PEG-3.35K, then imaged. Scale bar, 10 μ m.

6. In the paragraph describing the celastrol regulation on Nur77 phase separation, at line 251 the authors state that celastrol promotes not only the nuclear export of Nur77 but also its phase separation. This is not yet clear from the data shown at this point of the paper. Celastrol promotes the nuclear export of the protein and phase-separation could be simply an outcome of the protein accumulation in the cytosol. Part of the subsequent experiments proves that celastrol can directly mediate the interaction between proteins, supporting the hypothesis that is also regulating phase separation. Finally, from reading the manuscript it is unclear what K-80003 (in extended figure 4 it is referred to as K-8003) should do in this context compared to celastrol.

Response: Reviewer is concerned that data provided in the manuscript do not support our statement that “celastrol promotes not only the nuclear export of Nur77 but also its phase separation”. We made this statement because our real-time imaging of live cells revealed the appearance of Nur77 droplets emerging from the nuclear membrane. We agree with the reviewer that celastrol-induced cytoplasmic accumulation of Nur77 could be also due to cytoplasmic retention of Nur77 protein rather than its nuclear export. Thus, we have discussed both possibilities in the revised manuscript. Regarding K-80003, a RXR ligand (we apologize for the mislabeling in *Supplementary Fig. 4b in the original manuscript*)¹⁶, it was used as a negative control. We have now included a set of data on another Nur77 ligand, BI1071¹⁰, for comparison. In this case, we found that BI1071 could also induce cytoplasmic accumulation of Nur77 and its phase separation (**Figure 20**, *Fig. 4g in the revised manuscript*).

Figure 20. Nur77 ligand BI1071 promotes Nur77 phase separation.

Droplet formation of GFP-Nur77 and mutants in HeLa cells treated with the indicated compounds 1 hr (celastrol, 2 μ M; BI1071, 0.5 μ M; K-80003, 2 μ M). Left: Representative droplet images of transfected GFP-Nur77 and mutants. Enlarged view of inset is also shown. Scale bar, 10 μ m. Right panels, quantification of the diffusion and aggregation of GFP-Nur77 and mutants.

7. At lines 274-276, the authors suggest that mutation of Lys536 in Nur77 impairs the effect of celastrol on inducing their interaction with p62. Lys536 is involved in ubiquitination and ubiquitination is hypothesized to be involved in recruiting p62. However, it is not clear at this point that celastrol mediates this interaction or that is essential for the phase separation of these components. Indeed, mutations of the Lys does not impact the phase separation of p62/Nur77. Only in the subsequent lines the authors detail that Nur77-LBD has specific interaction with the p62 in presence of celastrol and that mutation of the Lys suppress the phase separation. This perhaps even suggests that the celestrol interactions is mediated by ubiquitin or that body celestrol and ubiquitin interactions are required for phase separation to occur. These observations are very important to define the multivalence of the states that are studied.

Response: Reviewer is concerned about the complex role of celastrol and ubiquitination in the phase separation of Nur77/p62. We apologize that we did not make it clear about our previous publication reporting that celastrol binding to Nur77 triggers Nur77 ubiquitination at Lys536 by inducing Nur77 interaction with TRAF2, an E3 ubiquitin ligase³. Lys536 is located at the C-terminal ligand-binding domain (LBD) of Nur77, and its ubiquitination is required to recruit p62 via its UBA domain. We described in the manuscript that mitochondrial Nur77 is ubiquitinated at Lys536 by TRAF2 and showed in Supplementary Fig. 5a that TRAF2 was recruited to the surface of Nur77 condensates in response to celastrol. We have now provided addition information regarding the effect of celastrol in promoting ubiquitin-dependent interaction between Nur77 and p62. Our observation that mutation of Lys536 abolished the interaction of p62 with Nur77-LBD but not with Nur77 (Fig. 5b and 5c in the original manuscript) led to our identification of another Nur77/p62 interaction mediated by the N-terminal IDR of Nur77 and the BP1 of p62. We agree that these observations are important to define the multivalent interaction between Nur77 and p62, which is triggered by celastrol binding. Nur77 is a ligand-dependent nuclear receptor. Our finding that a small molecule ligand can trigger a phase separation involving in mitophagy is highly significant in the field. We hope that the addition of schematic representations of domain structures of Nur77 and p62 and a graphic summary in the revised manuscript would help readers understanding the complex interactions between Nur77 and p62 and how each interaction contributes to the phase separation of Nur77/p62, leading to the sequestration of damaged mitochondria and their autophagy (**Figure 21**, *Fig 7e in the revised manuscript*).

Figure 21. Multivalent interaction between Nur77 and p62.

The interaction between Nur77 and p62 is mediated by the IDR of Nur77 and the PB1 of p62, which is celastrol-independent, whereas the interaction between LBD of Nur77 with the UBA of p62 depends on celastrol binding, which triggers the ubiquitination of Nur77-LBD.

8. At lines 180-186 as well as in the paper figures, FRAP data are reported in terms of half time and or percentage of recovery after a given lag time. This is sort of confusing, since it does not allow a one-to-one comparison. A “ $T(1/2)$ ” can be obtained also for measurements that do not recover completely and will represent the half-time of recovery of the dynamic part. The missing amplitude will represent the fraction of molecules not diffusing or diffusing on a much slower time scale. In this respect, instead of reporting a % of recovery after different lag times, it seems like it would be more significative to choose an identical lag time for all the measurements and compare the percentage of amplitude at that time.

Response: We agree with the suggestion, and have now used the percentage of recovery in 2 min for all the measurements (**Figure 21, Fig. 2d, 7g in the revised manuscript**).

Figure 21. FRAP analysis of p62 condensates formed with Nur77 or mutants.

FRAP analysis of the effect of Nur77 or mutant on p62 mobility after treatment with and/or without celastrol in HeLa cells. Scale bar, 1.5 μm .

9. At line 210 the authors states that they tested for whether the droplets were irreversible aggregates or reversible phase-separated condensates. The dilution and observation of smaller objects does not disprove the formation of irreversible or long-standing aggregates, since the smaller objects could well be the remaining aggregate. The proper test would consist in titrating decreasing concentrations of protein against a fixed crowder (e.g. PEG of a given size) concentration and identify the threshold for phase separation. Then take a solution that contains preformed droplet and dilute down the protein concentration (but not the polymer concentration) below the threshold concentration and show that there is no phase-separation. This partially performed at in extended figure 3d and, based on those observations, it would suffice to add 1 or 2 further dilutions in Ext.Fig. 3e to pass the threshold (about 0.1 μ M in PEG 3.35k in panel d, possibly not the same in PEG 8k panel e).

Response: We appreciate the insightful suggestion for our experiment characterizing the reversibility of Nur77 droplets (*Supplementary Fig. 3e in the original manuscript*). We have conducted the suggested experiments by gradient dilution of 12.5 μ M GFP-Nur77 protein in the 10% PEG-3.35K, and our new data showed that Nur77 condensates disappear completely (**Figure 22, Supplementary Fig. 3e in the revised manuscript**), again demonstrating that that Nur77 forms reversible condensates.

Figure 22. The formation of Nur77 droplets is reversible.

GFP-Nur77 (12.5 μ M) droplet were formed in 10% PEG-3.35K for 10 min, followed by a 1:5 dilution (diluted 1/5) in buffer containing PEG-3.35K (10%). The size and turbidity of GFP-Nur77 droplets was calculated and normalized (n = 4 experiments). Scale bar, 5 μ m.

10. In extended figures 1 and 2 the fluorescence intensities should have an adequate x-axis. Indication that the scale bar is 10 μ m (there is a typo in the legend saying 10 μ M) would provide an important information for comparison with other objects of analogous nature as well to enable comparison with simulations.

Response: x-axes for figures shown in Supplementary Fig. 1 and 2 have been modified as suggested.

11. The sentence at line 218-220 reads odd: “revealed that it exhibited as a single spherical particle”. It is not clear the subject of the sentence and the objects do not appear to be perfectly spherical.

Response: We have revised the sentence to “3D-reconstruction of the GFP-Nur77 revealed that it exhibited as a single particle”.

12. Similarly, the sentence at 227-228, “we found that... displays the highest disorder tendency in a broader IRD lacking well-defined structure” can perhaps be simplified to be “we found that... exhibits the highest structural disorder propensity”.

Response: We have modified the sentence as suggested.

References

1. Hunt NJ, Kang SWS, Lockwood GP, Le Couteur DG, Cogger VC. Hallmarks of Aging in the Liver. *Comput Struct Biotechnol J* **17**, 1151-1161 (2019).
2. Chen G, Kroemer G, Kepp O. Mitophagy: An Emerging Role in Aging and Age-Associated Diseases. *Front Cell Dev Biol* **8**, 200 (2020).
3. Hu MJ, *et al.* Celastrol-Induced Nur77 Interaction with TRAF2 Alleviates Inflammation by Promoting Mitochondrial Ubiquitination and Autophagy. *Molecular Cell* **66**, 141-+ (2017).
4. Horvath S, *et al.* Obesity accelerates epigenetic aging of human liver. *Proc Natl Acad Sci U S A* **111**, 15538-15543 (2014).
5. Wakabayashi T. Megamitochondria formation - physiology and pathology. *J Cell Mol Med* **6**, 497-538 (2002).
6. Yamada T, *et al.* Mitochondrial Stasis Reveals p62-Mediated Ubiquitination in Parkin-Independent Mitophagy and Mitigates Nonalcoholic Fatty Liver Disease. *Cell Metab* **28**, 588-604 e585 (2018).
7. Sanchez-Martin P, Sou YS, Kageyama S, Koike M, Waguri S, Komatsu M. NBR1-mediated p62-liquid droplets enhance the Keap1-Nrf2 system. *EMBO Rep* **21**, e48902 (2020).
8. Corson TW, Crews CM. Molecular understanding and modern application of traditional medicines: triumphs and trials. *Cell* **130**, 769-774 (2007).
9. Liu J, Lee J, Salazar Hernandez MA, Mazitschek R, Ozcan U. Treatment of obesity with celastrol. *Cell* **161**, 999-1011 (2015).
10. Chen XH, *et al.* BI1071, a Novel Nur77 Modulator, Induces Apoptosis of Cancer Cells by Activating the Nur77-Bcl-2 Apoptotic Pathway. *Mol Cancer Ther* **18**, 886-899 (2019).
11. Wurzer B, *et al.* Oligomerization of p62 allows for selection of ubiquitinated cargo and isolation membrane during selective autophagy. *Elife* **4**, e08941 (2015).
12. Sun D, Wu R, Zheng J, Li P, Yu L. Polyubiquitin chain-induced p62 phase separation drives autophagic cargo segregation. *Cell Res* **28**, 405-415 (2018).
13. Zaffagnini G, *et al.* p62 filaments capture and present ubiquitinated cargos for autophagy. *EMBO J* **37**, (2018).

14. Mitrea DM, *et al.* Nucleophosmin integrates within the nucleolus via multi-modal interactions with proteins displaying R-rich linear motifs and rRNA. *Elife* **5**, (2016).
15. Lefebvre P, Benomar Y, Staels B. Retinoid X receptors: common heterodimerization partners with distinct functions. *Trends Endocrin Met* **21**, 676-683 (2010).
16. Chen L, *et al.* Modulation of nongenomic activation of PI3K signalling by tetramerization of N-terminally-cleaved RXRalpha. *Nat Commun* **8**, 16066 (2017).
17. Wansa KD, Harris JM, Muscat GE. The activation function-1 domain of Nur77/NR4A1 mediates trans-activation, cell specificity, and coactivator recruitment. *J Biol Chem* **277**, 33001-33011 (2002).

Reviewers' Comments:

Reviewer #1:

Remarks to the Author:

The revised manuscript improved considerably, and it is well worth publishing the revision in Nature Communications. Congratulations. I had one minor comment. In Figure 4a, the authors should show the blot for LC3 to prove the blockade of autophagy.

Reviewer #2:

Remarks to the Author:

The authors have satisfactorily addressed most of my concerns. They have improved their data presentation and included new figures. However, before I would recommended acceptance of this manuscript I do have a few minor comments.

1) Supplementary figure 1. Why was next to Celastrol, TNFa added as a stimulus? As immediate-early gene, Nur77 expression is rapidly induced upon administration of TNFa. How do you know that all the stimulations shown in suppl. 1 are due to stimulation with Celastrol and not because Nur77 expression is increased as response to TNFa? Did you also perform experiments with Celastrol alone, or TNFa alone? Figure 1b of the original manuscript does not indicate TNFa, while the exact same figure suppl 1d in the revised manuscript does indicate this. The authors should provide the reasoning why they have added TNFa and also describe better that the effects observed could be due to TNFa. Was TNFa also used in the other experiments shown throughout the manuscript?

2) The authors now present the plots of the gating strategy. Line 733 states that "single and living cells was gated by P1". By just gating SSC-A vs FSC-A it is possible to exclude the (cellular) debris, but not the doublets.

3) I may have missed the data, but the authors state that they have conducted a color swab to exclude diffraction effects when comparing the diameter of de bodies of p62, Nur77 and L3, but I do not see the data. Also because they now present different results (bodies of p62, Nur77 and LC3 have a greater diameter than those of mitochondria (in the revised version) vs the condensates of p62 and LC3 had greater diameter compared with those of Nur77 and mitochondria (in the original manuscript)), these experiments thus show quite some variability. If the authors do want to claim these findings, do authors need to show their color swab data.

4) Suppl figure 3c; the reviewers have repeated their experiments and present new data to show spherical instead of amorphous Nur77 droplets in 30% PEG-3,35K and at 5uM concentration. In the revised version they present the same quantification plots for these data as in the original manuscript, while the data look different. Did the authors do new calculations do calculate the size and turbidity on the new data?

Reviewer #3:

Remarks to the Author:

The authors have addressed all my concerns.

Reviewer #4:

Remarks to the Author:

This is a revised manuscript. The authors have done an excellent job to address previous reviewers' critiques and provided additional experimental data. This reviewer would agree with the previous reviewers' assessment that this is an interesting and novel study, which may have broad significance on phase-separation and mitophagy field. However, this reviewer did have some concerns on the mitophagy data interpretation. The heavily (actually only) rely on the use of a

single EGFP-mCherry-COX8 assay for mitophagy is generally fine but perhaps more evidence for mitophagy (autolysosome/lysosome-mediated mitochondria degradation) should be provided.

Specific comments:

1. This reviewer was confused by using the flow cytometry analysis for EGFP-mCherry-COX8. While it is easily to distinguish Red-only puncta of COX8 as in the acidic compartments (presumably in the lysosomes) under confocal fluorescence microscopy, authors used flow cytometry to measure all the red-channel (PE) signals that would cover all the "yellow+red" mitochondria. However, if the cell number is the same, all the "yellow" mitochondria also have red fluorescence! Unless authors clearly demonstrated that the "red fluorescence intensity" is higher in lysosome compartment than the cytosolic mitochondria. This further support the concern and justify the need for another additional mitophagy marker such as the levels of inner mitochondrial membrane proteins by western blot etc.
2. Figure 1C, the 10 nm and 15 nm gold particles are too close to see the difference. It was not clear why ultras-small gold particles such as 5 nm was not used?
3. Figure 1g, authors only showed one cell, how did this represent the liver? Liver has unique zonation nation, at least some liver structure should be shown? Any mitophagy activity difference in different liver zonation? Low magnification images are needed to show multiple cells, ideally including the unique liver portal vein or central vein (the LC3 and HSP60 staining in the liver is much better). Were these mice fasted when the liver tissues were collected?

Response to Reviewer #1

Reviewer #1 (Remarks to the Author):

The revised manuscript improved considerably, and it is well worth publishing the revision in Nature Communications. Congratulations. I had one minor comment. In Figure 4a, the authors should show the blot for LC3 to prove the blockade of autophagy.

Response: We appreciate Reviewer's time and effort in evaluating our manuscript. A minor concern remaining relates to an additional control for autophagy-deficient cells that we used to evaluate the mitophagic effect of celastrol (Figure 4a in our previous response to Reviewer 1). As suggested, we have included a new immunoblot showing reduced expression of LC3-II expression in autophagic deficient cells in the absence or presence of celastrol. Our data also demonstrated that celastrol-induced downregulation of COXII (mitochondrial inner membrane protein), an indicative of the degradation of mitochondria, was attenuated in autophagy-deficient cells (**Figure 1**). These new results are included in Supplementary Fig. 1d in the revised manuscript. We hope that these additional data address Reviewer's concern.

Figure 1: Characterization of Atg7-deficient cells. Induction of LC3II and reduction of COXII expression by celastrol were abrogated in Atg7-deficient cells. HeLa cells transfected with control or Atg7 siRNA were treated with or without celastrol (2 μ M) and TNF α (20 ng/ml) for 12 hr, and analyzed by Western blot.

Response to Reviewer #2

We thank the Reviewer for evaluating our manuscript, and appreciate the positive comments about the data that we provided to address Reviewer's previous concerns. There are some minor concerns remaining, including the role of TNF α in celastrol-induced mitophagy, the gating strategy of our flow cytometry experiments, and the color swab data. Our detailed responses to the Reviewer's concerns follow (Reviewer's comments are italicized).

The authors have satisfactorily addressed most of my concerns. They have improved their data presentation and included new figures. However, before I would recommend acceptance of this manuscript I do have a few minor comments.

1) Supplementary figure 1. Why was next to Celastrol, TNF α added as a stimulus? As immediate-early gene, Nur77 expression is rapidly induced upon administration of TNF α . How do you know that all the stimulations shown in suppl. 1 are due to stimulation with Celastrol and not because Nur77 expression is increased as response to TNF α ? Did you also perform experiments with Celastrol alone, or TNF α alone? Figure 1b of the original manuscript does not indicate TNF α , while the exact same figure suppl 1d in the revised manuscript does indicate this. The authors should provide the reasoning why they have added TNF α and also describe better that the effects observed could be due to TNF α . Was TNF α also used in the other experiments shown throughout the manuscript?

Response: Reviewer is concerned that we added TNF α as a stimulus in Supplementary Figure 1a. We used TNF α to trigger inflammatory signaling, which are known to insult mitochondria, making them prone to autophagy. This is a condition that we described previously to stimulate mitophagic effect of celastrol (*Mol Cell* 66, 141, 2017, Ref 28 in the revised manuscript)¹. As we considered this is a published condition to induce mitophagy, we did not put TNF α in the labeling of Supplementary Figure 1 in our original submitted manuscript, although we clearly described that TNF α was used in combination with celastrol in the figure legend. However, Reviewer 1 in his/her critique 9 questioned whether this was mislabeled. To address Reviewer 1's concern, we had in our revised manuscript included TNF α in our labeling. We sorry for this confusion. TNF α was used together with celastrol to induce mitophagy throughout the manuscript.

Regarding the role of TNF α in celastrol-induced mitophagy, we had extensively characterized its role in celastrol-induced mitophagy in our previous study (*Mol Cell* 66, 141, 2017)¹. In inflammatory animal models, administration of animals with celastrol was sufficient to induce mitophagy. However, TNF α alone can not induce mitophagy but it can potentiate the mitophagic effect of celastrol in cell culture. Mitochondria are susceptible to inflammatory signaling, which can result in accumulation of inflammatory mediators at dysfunctional mitochondria. We and others have shown that mitochondria are dysfunctional when cells are treated with TNF α , which is accompanied with increased accumulation of tumor necrosis factor receptor-associated factor 2 (TRAF2) at damaged mitochondria (*Mol Cell* 66, 141, 2017; *Circ Heart Fail* 8, 175, 2015; *Cell Death Differ* 17, 1420, 2010)^{1, 2, 3}. We then found that

mitochondrial TRAF2 could interact with mitochondrial Nur77 in the presence of celastrol. The interaction of mitochondrial TRAF2 (an E3 ligase) and Nur77 led to ubiquitination of Nur77, thereby priming dysfunctional mitochondria for autophagy. We had therefore attributed the role of TNF α largely to its inflammatory insult to mitochondria, a condition that promotes dysfunctional mitochondria for autophagic degradation by recruiting TRAF2. As these data have been published¹, we did not discuss them in detail in the current manuscript. In the revised manuscript, we have described the role of TNF α in celastrol-induced mitophagy to make it clear to readers.

Reviewer is correct that TNF α could induce the expression of Nur77. However, TNF α -induced Nur77 remains in the nucleus, whereas celastrol acts to promote the translocation of Nur77 from the nucleus to mitochondria and the interaction of Nur77 and TRAF2 at dysfunctional mitochondria. TNF α -induced Nur77 expression may contribute to the mitophagic effect of celastrol as celastrol can induce the translocation of TNF α -induced Nur77 from the nucleus to dysfunctional mitochondria to mediate their degradation.

2) The authors now present the plots of the gating strategy. Line 733 states that “single and living cells was gated by P1”. By just gating SSC-A vs FSC-A it is possible to exclude the (cellular) debris, but not the doublets.

Response: Reviewer is concerned about our gating strategy that can not exclude the doublets. We fully agree with the comment. Accordingly, we have re-analyzed our flow cytometry data by using forward scatter height (FSC-H) and forward scatter area (FSC-A) density plot (gate: Single Cells) to exclude doublets (**Figure 1**). Old flow cytometry data have been replaced with our newly obtained results (Supplementary Figure 1 and 6). The new gating strategy is also described in Supplementary Figure 1a in the revised manuscript.

Figure 1. Gate setting strategy of FACS plots of EGFP-mCherry-COX8 assay. EGFP-mCherry-COX8 positive cells were first gated (gate: P1) based on forward scatter area (FSC-A) and side scatter area (SSC-A), followed by doublet discrimination based on FSC height (FSC-H) by FSC-A (gate: Single Cells). EGFP-mCherry-COX8 positive cells were determined using untransfected controls (gate: FITC). Measurements EGFP-mCherry-COX8 were made using dual-excitation ratiometric pH measurements at FITC (488 nm, pH 7) and PE (561 nm, pH 4) lasers respectively. For each sample, 10,000 events of FITC positive cells were collected and subsequently gated with appropriate controls to detect mCherry-COX8 exposure cells but not EGFP-mCherry-COX8 double-positive cells.

3) I may have missed the data, but the authors state that they have conducted a color swab to exclude diffraction effects when comparing the diameter of the bodies of p62, Nur77 and LC3, but I do not see the data. Also because they now present different results (bodies of p62, Nur77 and LC3 have a greater diameter than those of mitochondria (in the revised version) vs the condensates of p62 and LC3 had greater diameter compared with those of Nur77 and mitochondria (in the original manuscript)), these experiments thus show quite some variability. If the authors do want to claim these findings, do authors need to show their color swab data.

Response: We apologize for the insufficient description of the suggested color swab data shown in Supplementary Fig. 2c and 2d. We agree with the reviewer that comparing the diameters of different condensates using immunofluorescence results can generate some variability. Thus, we have now eliminated our data showing the comparison of the diameter of each condensate, and show only the colocalization of p62, Nur77, LC3 and mitochondria in cells treated with celastrol (**Figure 2**, Supplementary Fig. 2c and 2d in the revised manuscript). Such a rearrangement will not affect the main conclusions drawn from our manuscript.

Figure 2. Characterization of Nur77/p62 condensates. (A) Celastrol-induced Nur77 condensates colocalize with mitochondria and p62. HeLa cells transfected with GFP-Nur77 and mCherry-p62 were treated with or without celastrol (2 μ M) and TNF α (20 ng/ml) for 1 h, and examined by confocal microscopy. (B) Celastrol-induced Nur77 condensates colocalize with mitochondria and LC3. HeLa cells transfected with GFP-Nur77 and mCherry-LC3 were treated with or without celastrol and TNF α , and examined by confocal microscopy. The fluorescence intensity was analyzed. Scale bar, 10 μ m.

4) Suppl figure 3c; the reviewers have repeated their experiments and present new data to show spherical instead of amorphous Nur77 droplets in 30% PEG-3,35K and at 5 μ M concentration. In the revised version they present the same quantification plots for these data as in the original manuscript, while the data look different. Did the authors do new calculations do calculate the size and turbidity on the new data?

Response: Reviewer is concerned about our data shown in Supplementary Figure 3c. We apologize that we did not replace the old quantitation plots with new data. The old quantitation plots are now replaced with new ones (**Figure 3**). Thanks for the careful examination of our manuscript, which allows us to correct the mistake!

Figure 3. Effect of PEG concentration on Nur77 droplet formation. Concentration dependent effect of PEG-3.35K on mCherry-Nur77 droplet formation. Left: Representative images of mCherry-Nur77 droplet at the indicated PEG-3.35K concentration. Right: The size and turbidity of droplets was calculated and normalized (n = 3). Scale bar, 5 μ m. Scale bar, 5 μ m.

References:

1. Hu MJ, *et al.* Celastrol-Induced Nur77 Interaction with TRAF2 Alleviates Inflammation by Promoting Mitochondrial Ubiquitination and Autophagy. *Molecular Cell* **66**, 141-+ (2017).
2. Yang KC, *et al.* Tumor necrosis factor receptor-associated factor 2 mediates mitochondrial autophagy. *Circ Heart Fail* **8**, 175-187 (2015).
3. Kim JJ, Lee SB, Park JK, Yoo YD. TNF-alpha- induced ROS production triggering apoptosis is directly linked to Romo1 and Bcl-X-L. *Cell Death Differ* **17**, 1420-1434 (2010).

Response to Reviewer #4

We thank the Reviewer for evaluating our manuscript, and appreciate the positive comments about our revised manuscript. The major concerns of Reviewer 4 relate to our mitophagy data interpretation and liver data presented. As suggested, we have provided several new sets of data to illustrate the feasibility of using FACS-based EGFP-mCherry-COX8 mitophagy assay and low magnification images of liver tissues so that some liver structures could be seen. Our detailed responses to the Reviewer's concerns follow (Reviewer's comments are italicized).

This is a revised manuscript. The authors have done an excellent job to address previous reviewers' critiques and provided additional experimental data. This reviewer would agree with the previous reviewers' assessment that this is an interesting and novel study, which may have broad significance on phase-separation and mitophagy field. However, this reviewer did have some concerns on the mitophagy data interpretation. The heavily (actually only) rely on the use of a single EGFP-mCherry-COX8 assay for mitophagy is generally fine but perhaps more evidence for mitophagy (autolysosome/lysosome-mediated mitochondria degradation) should be provided.

Specific comments:

1. This reviewer was confused by using the flow cytometry analysis for EGFP-mCherry-COX8. While it is easily to distinguish Red-only puncta of COX8 as in the acidic compartments (presumably in the lysosomes) under confocal fluorescence microscopy, authors used flow cytometry to measure all the red-channel (PE) signals that would cover all the "yellow+red" mitochondria. However, if the cell number is the same, all the "yellow" mitochondria also have red fluorescence! Unless authors clearly demonstrated that the "red fluorescence intensity" is higher in lysosome compartment than the cytosolic mitochondria. This further support the concern and justify the need for another additional mitophagy marker such as the levels of inner mitochondrial membrane proteins by western blot etc.

Response: Reviewer is concerned about flow cytometry analysis of EGFP-mCherry-COX8 reporter. Using FACS-based approach to quantitate mitophagy has been increasingly used to study mitophagy. Our FACS-based mitophagy assay follows the protocol described by Lazarou et al. (*Nature* 524, 309, 2015)¹ using dual excitation ratiometric pH measurements at 488 (pH 7) and 561 (pH 4) nm lasers to measure the engulfment of mitochondrial EGFP-mCherry-COX8 reporter into lysosome. We set Y-axis as COX8 PH 4 (PE) and X-axis as COX8 PH 7 (FITC) with appropriate controls to detect spectral shift due to the engulfment of mitochondrial EGFP-mCherry-COX8 reporter into lysosome owing to low pH.

We agree with the reviewer that more data are needed to validate the FACS-based strategy. Therefore, several sets of data are provided to support the feasibility of using FACS-based EGFP-mCherry-COX8 mitophagy assay. As suggested, we first conducted several experiments to demonstrate that the red fluorescence intensity of EGFP-mCherry-COX8 reporter is higher in lysosome compartment than the cytosolic mitochondria. We transfected EGFP-mCherry-COX8 reporter plasmid into HeLa cells, which were then treated with celastrol and TNF α , a

condition that is known to induce mitophagy (*Mol Cell* 66, 141, 2017)². Confocal microscopy analysis revealed two different types of mitochondrial morphology, puncta and network, both of which displayed red fluorescence (**Figure 1A**). The intensity of red fluorescence displayed by mitochondrial puncta was much higher than that shown by mitochondrial network. When merged, the red image of mitochondrial network colocalized with green mitochondrial network, displaying yellow image. This resulted in red-only mitochondrial puncta. To study whether red-only mitochondrial puncta represent mitochondria undergoing autophagy, we also stained cells with lyso-tracker and found that they were indeed in lysosome (**Figure 1B**). Thus, these results demonstrated that mitochondria identified by red-only puncta of COX8 undergo autophagy in lysosome, thereby demonstrating the feasibility of using FACS-based EGFP-mCherry-COX8 mitophagy assay. The assay is very sensitive as the red-only puncta of COX8 were also detected in liver tissue of young mice (**Figure 1C**).

Figure 1. Characterization of red fluorescence intensity of EGFP-mCherry-COX8 in cells undergoing mitophagy. (A) HeLa cells transfected with EGFP-mCherry-COX8 were treated with or without celastrol (2 μM) and TNFα (20 ng/ml) for 1 h. Fluorescence of EGFP-mCherry-COX8 was analyzed by using 488 nm and 561 nm laser. The white arrow indicates the fluorescence intensity analysis showing that the red fluorescence of mitochondrial puncta (black dotted box) was stronger than that of mitochondrial network (purple dotted box). Scale bar, 10 μm. (B) HeLa cells transfected with EGFP-mCherry-COX8 were treated with or without celastrol/TNFα, and lysosomes were stained by lysosome tracker. The white arrow indicates the fluorescence intensity analysis showing that the red puncta were co-localized with lysosomes. The black dotted box shows mitophagy, and purple dotted box shows mitochondria. Scale bar, 10μm. (C). Fluorescence intensity analysis of EGFP-mCherry-

COX8 in the liver from wild-type young mice. The white arrow indicates the fluorescence intensity analysis showing that the red fluorescence of autophagic mitochondria (black dotted box) was stronger than that of normal mitochondria (purple dotted box). Scale bar, 2 μ m.

As suggested, we also examined the levels of inner mitochondrial membrane proteins by Western blot to further confirm the mitophagic effect of celastrol. Thus, HeLa cells treated with or without celastrol/TNF α were evaluated for the expression levels of cytochrome *c* oxidase subunit II (COXII), a mitochondrial DNA (mtDNA)-encoded inner membrane protein (**Figure 2A-C**). Our results demonstrated that the protein levels of COXII were significantly reduced upon celastrol/TNF α treatment, revealing the mitophagic effect of celastrol/TNF α treatment. Furthermore, down-regulating the expression of Nur77, p62, or Atg7 by siRNA transfection attenuated the mitophagic effect of celastrol/TNF α . These results are in agreement with other data reported in this manuscript and our previous results (*Mol Cell* 66, 141, 2017)² showing that Nur77 and p62 are required for the mitophagic effect of celastrol.

To substantiate the mitophagic effect of different Nur77 mutants including Nur77-K536R, Nur77-IDR, Nur77- Δ IDR, and Nur77-LBD used in the manuscript, we transfected them into Nur77-depleted HeLa cells (Nur77^{-/-}HeLa) and evaluated their role in supporting the mitophagic effect of celastrol (**Figure 2D**). Our results showed that transfection of Nur77 but not any other mutants could rescue the effect of celastrol on inducing COXII degradation in Nur77^{-/-}HeLa cells. These results are consistent with those obtained by FACS-based EGFP-mCherry-COX8 mitophagy assay (Supplementary Figure 6a) and are included in the revised manuscript (Supplementary Figure 6b) to support our conclusion regarding the mitophagic role of Nur77 mutants.

Together, our new data provide additional evidence to support the use of FACS-based EGFP-mCherry-COX8 mitophagy assay and the conclusion made in the manuscript.

Figure 2. Characterization of celastrol-induced mitophagy. (A-C) Celastrol-induced degradation of mitochondrial membrane proteins depends on the expression of Nur77, p62, and Atg7. HeLa cells transfected with or without the indicated siRNA were treated with or without celastrol (2 μ M) and TNF α (20 ng/ml) for 12 h, and analyzed by Western blot. (D) Characterization of the mitophagic effect of

Nur77 mutants. HeLa cells depleted with Nur77 (Nur77^{-/-} HeLa) were transfected with Nur77 and mutants (Nur77-K536R, Nur77-IDR, Nur77-ΔIDR, and Nur77-LBD), treated with or without celastrol and TNFα, and analyzed by Western blot.

2. Figure 1C, the 10 nm and 15 nm gold particles are too close to see the difference. It was not clear why ultrasmall gold particles such as 5 nm was not used?

Response: Regarding the concern that why not use 5 nm gold particles in Figure 1C, we were afraid that the 5 nm gold particles might be too small to distinguish them from background particles. To address the concern that the 10 nm and 15 nm gold particles are too close to see the difference, we have now revised the figure by including zoom insets of gold particles (**Figure 3**), which now clearly show the difference between 10 nm and 15 nm gold particles.

Figure 3. Nur77 and p62 colocalize with the mitochondrial outer membrane during celastrol-induced Mitophagy. (A) Electron micrographs of HeLa cells stained with 15 nm immunogold-conjugated Nur77 antibody to detect Nur77 (red), and 10 nm immunogold-conjugated p62 antibody to detect p62 (green). Cells were treated for 1 h with celastrol (2 μM) and TNFα (20 ng/ml). Blue dotted line indicates autophagosome. Green dotted line indicates zoom inset of immunogold. (B) Electron micrographs of HeLa cells stained with 15 nm immunogold-conjugated LC3 antibody to detect LC3, and 10 nm immunogold-conjugated p62 antibody to detect p62. Cells were treated with celastrol and TNFα. The blue dotted line indicates autophagosome. The green dotted line indicates a zoom inset of immunogold. Mito, mitochondrion, Scale bar, 200 nm.

3. Figure 1g, authors only showed one cell, how did this represent the liver? Liver has unique zonation, at least some liver structure should be shown? Any mitophagy activity difference in different liver zonation? Low magnification images are needed to show multiple cells, ideally including the unique liver portal vein or central vein (the LC3 and HSP60

staining in the liver is much better). Were these mice fasted when the liver tissues were collected?

Response: Reviewer is concerned that only one cell was shown in Figure 1g. We fully agree with the concern, and have provided low magnification images showing multiple cells from liver tissues as suggested (**Figure 4A**). All tissue specimens were obtained from the middle region of the large lobe of the liver, in which liver structure including the portal vein (PV) or central vein (CV) could be seen. These data are now included as Supplementary Fig. 1q. We have also conducted experiments to examine mitophagy in different liver zonation and detected mitophagy in three different zones in wild-type young mice (**Figure 4B**). The significance of mitophagy in different zone remains to be investigated. Regarding whether mice were fasted when tissues were collected, they were on regular diet when the liver tissues were collected to avoid the effect of autophagy induced by starvation.

Figure 4. Nur77 is necessary for liver mitophagy. (A) Representative images of EGFP-mCherry-COX8 shows mitochondrial autophagy in the liver from aging model. (B) Representative images show mitochondrial autophagy from the different zone of liver. Scale bar, 30 μ m. Central vein (CV) and portal vein (PV) are indicated.

Reference:

1. Lazarou M, *et al.* The ubiquitin kinase PINK1 recruits autophagy receptors to induce mitophagy. *Nature* **524**, 309-314 (2015).
2. Hu MJ, *et al.* Celastrol-Induced Nur77 Interaction with TRAF2 Alleviates Inflammation by Promoting Mitochondrial Ubiquitination and Autophagy. *Molecular Cell* **66**, 141-+ (2017).

Reviewers' Comments:

Reviewer #1:

Remarks to the Author:

The authors have adequately addressed my concern.

Reviewer #2:

Remarks to the Author:

The authors have satisfactorily addressed my remaining concerns and I now fully support acceptance of this work in Nature Communications.

Reviewer #4:

Remarks to the Author:

The authors did an excellent job for the revision and provided new data. My concerns have been adequately addressed.

Response to Reviewer #1

Remarks to the Author: *The authors have adequately addressed my concern.*

Response: Thanks!

Response to Reviewer #2

Remarks to the Author: *The authors have satisfactorily addressed my remaining concerns and I now fully support acceptance of this work in Nature Communications.*

Response: Thanks!

Response to Reviewer #4

Remarks to the Author: *The authors did an excellent job for the revision and provided new data. My concerns have been adequately addressed.*

Response: Thanks!